# A method for characterizing the spatial organization of deep convective cores in deep convective systems' cloud shield using idealized elementary convective structure decomposition

**Louis Netz[1], Thomas Fiolleau[1] and Rémy Roca[1]**

[1] Université de Toulouse, Laboratoire d'Etudes en Géophysique et Océanographie Spatiales (CNRS/CNES/IRD/UPS), Toulouse, France

Corresponding author: Louis Netz (louis.netz@cnrs.fr)

**ORCID: Louis Netz 0009-0009-2112-8453**

**ORCID: Thomas Fiolleau 0000-0001-5902-1701**

**ORCID: Rémy Roca 0000-0003-1843-0204**

**Abstract**

Deep convective systems (DCSs) play a fundamental role in atmospheric dynamics, precipitation, cloud radiative effects, and large-scale circulations. Their associated deep convection exhibits complex spatial arrangements, commonly referred to as convective organization, which exerts an influence on the systems' morphology that needs to be assessed. However, quantifying this organization remains challenging due to the lack of a general, robust, and consensual metric, in both observations and models. This study introduces a new method for characterizing the spatial arrangement of deep convective cores within the cloud shield of individual DCSs. The first step of this technique consists in decomposing the convective mask into elementary structures. Four key parameters are then extracted to fully capture the organization of a scene. Two of these parameters characterize the overall properties of the convective field, such as the size and convective fraction of convective cores. The remaining two parameters are specifically designed to describe the spatial arrangement of deep convective cores: a characteristic convective scale using two-dimensional (2D) autocorrelation and an evaluation of the deviation from randomness by comparing it to a stochastic ensemble of synthetic convective fields. Two independent datasets, derived from satellite observations and idealized kilometer-scale numerical simulations, each employing distinct convective core identification techniques are used to assess the generalization of the method. Finally, an unsupervised clustering algorithm identifies four distinct classes, revealing consistent and physically sound patterns of convective organization across both datasets. This demonstrates the method's robustness and suitability to characterize the spatial organization of convective cores in convective systems' cloud shield.

**Short summary:**

Large convective systems are the primary drivers of the Earth's rainfall and climate, yet the spatial organization of their associated convection remains poorly described. This study presents a straightforward approach to characterizing this organization. First, the convective field is decomposed into elementary structures, and then four scores are computed to describe cores' size, density, spacing

scale, and departure from randomness. Applied to both satellite data and km-scale simulations, the
method robustly yields the same overall organization characterization.

## 1 Introduction

In the atmosphere, deep convection occurs mainly in the form of structures where multiple individual
convective cells are spatially organized yielding to a wide diversity of convective cloud systems, ranging
from hurricanes to squall lines (Lafore et al. 2017). The largest and longest lived of these systems are often
termed Mesoscale Convective System and are also characterized by a specific organization of deep
convection inside the cloud system's shield (Houze 2004). Yet, the reasons for this organization remain
challenging to assess (Muller et al. 2022). Large scale forcing is often invoked (Markowski and Richardson
2010) and cell to cell interaction (Mapes 1993) is also a process identified to explain the self-organization
of deep convection in the absence of large scale forcing as observed in the tropics (LeMone, Zipser, and
Trier 1998; Holloway et al. 2017).  Recent investigation suggests radiative feedback between the cloud
and convection also contribute to the organization of deep convection (Muller and Bony 2015; Wing and
Emanuel 2014). The consolidation of our physical understanding of the role of deep convection
organization is nevertheless hampered by the lack of a univocal definition of organization (Retsch, Jakob,
and Singh 2020).
Indeed, if anything organization is a loose term. Encompassing descriptive perspective (Gallus, Snook, and
Johnson 2008) to mathematical formulation (e.g., Tobin, Bony, and Roca 2012), organization refers to
various spatial arrangements. Departure of such arrangement from randomly arranged low level cloud
fields has been a starting point of numerous investigations that prompted complementary concepts like
clustering and regularity (Weger et al. 1992). Recently, these concepts were integrated into a single metric
that could discern regular, random, and clustered cloud scenes (Biagioli and Tompkins 2023; Tompkins
and Semie 2017). The relevance of such indices to the specificity of deep convection within a convective
cloud shield is not straightforward. First, owing to its formulation, a minimum number of convective cores
is required to compute the metrics (Mandorli and Stubenrauch 2023); a criterion which is not easily
satisfied for deep convective systems (Schiro et al. 2020). Second, most of the metrics are computed over
a given regular area in contrast with the variable nature of the deep convective systems cloud shield area
(Roca, Fiolleau, and Bouniol 2017). Finally, another source of difficulty arises from the very definition of
deep convection, be it point wise or object oriented (Takahashi et al. 2023). Other existing metrics that
are more object oriented such as COP, ABCOP and ROME (White et al. (2018), Jin et al. (2022), Retsch et
al. (2020)) consider different convection parameters, like the object's area, mean area of objects, relative
or geometric distance between objects, in addition to their center within a specific grid.
Building on previous efforts and acknowledging their respective strengths and limitations, we introduce a
novel approach that employs an algorithm to compute organizational metrics of deep convective cores.
These computations are conducted across multiple scenes characterized by varying extents of DCSs' cloud
shield. This ensemble of scenes is then classified to reveal four, well separated, unambiguous classes of
organization. The procedure is performed on two datasets using different identification of deep
convection, a hydrometeor and a dynamical perspective, to assess the sensitivity of the overall method
to the definition of deep convection. The article is articulated as follows. Section 2 introduces the data
and the convective core identification. Section 3 summarizes the algorithm and its implementation. A
summary and discussion are offered in section 4 and a conclusion section ends the paper.

## 2 Data

### 2.1 Satellite observations

#### Georing Infrared Observations

Thermal infrared brightness temperature data from the operational meteorological geostationary satellite fleet (GEOring) were used to monitor deep convective systems across the tropical belt throughout the 2012–2020 period. As highlighted by Fiolleau et al. (2020), the GEOring is not a homogeneous suite of instruments operating under uniform conditions. Differences in spatial and temporal resolution, spectral filter functions, and calibration procedures vary between platforms, leading to systematic biases in brightness temperature measurements. To address these discrepancies, Fiolleau et al. (2020) homogenized thermal infrared data from the geostationary satellite fleet for cold cloud studies, using the Scanner for Radiation Budget (ScaRaB) IR channel onboard Megha-Tropiques as a reference. The resulting GEOring IR dataset was inter-calibrated, spectrally adjusted, and corrected for limb darkening over a nine-year period.

To complete the harmonization process, the temporal resolution was standardized to 30 minutes across the GEOring, and all geostationary data were remapped onto a common 0.04° equal-angle longitude-latitude grid (Fiolleau et al., 2020). The spatial coverage of each geostationary platform was selected to provide substantial overlap with adjacent satellites, ensuring seamless data continuity between 55°S and 55°N. The fully homogenized IR GEOring dataset is comprehensively described in Fiolleau et al. (2020) and Fiolleau & Roca (2024).

#### Convective System cold cloud shield from the TOOCAN algorithm

The TOOCAN algorithm is a 3D segmentation-based method designed to track Deep Convective Systems (DCSs) using geostationary infrared (IR) satellite imagery. It identifies DCSs within a time series of IR images through a spatio-temporal region-growing technique, which iteratively detects and expands convective seeds. If present, convective seeds are first identified at 190 K, requiring a minimum lifetime of 1h30 (3 frames) and an area of at least 625 km² per frame. These are then dilated using a 10-connected spatiotemporal neighborhood operator until reaching a boundary that is 2 K warmer. A second convective seed detection is subsequently applied at 192 K. This iterative process continues, progressively expanding the identified seeds and detecting new ones at every 2 K increment, until reaching the 235 K boundary (Fiolleau and Roca, 2013).

The algorithm decomposes the high cold cloud shield below 235 K into multiple DCSs, even when their anvil clouds are interconnected. This approach enables a comprehensive identification of DCSs, ranging from small, short-lived, isolated systems to long-lived, extensive convective systems that can propagate over several hundred kilometers and span several thousand square kilometers. More importantly, TOOCAN suppresses split and merge artefacts, a key limitation of traditional overlap-based tracking methods. By capturing the full spectrum of convective system organization, TOOCAN improves the representation of convective evolution, offering a more reliable characterization of deep convective systems (DCSs).

In this study, we use a database of deep convective systems and their morphological characteristics covering the 2012–2020 period over the intertropical belt, fully described in Fiolleau and Roca (2024). TOOCAN has been systematically applied to the homogenized GEOring IR data over this period. The resulting database provides access to key morphological parameters of each DCS, including its location and time of initiation and dissipation, lifetime duration, propagation distance, and maximum cold cloud

extent. Additionally, the dataset documents the evolution of morphological properties throughout the
DCS life cycle. A total of $15 \times 10^6$ DCSs have been detected and tracked by TOOCAN across tropical regions
during this nine-year period.

## Deep convective mask from collocated radar measurements

IR-only data from geostationary satellites are limited to the detection of cloud-top characteristics and do
not provide direct information on the internal processes, vertical structure, or dynamical evolution of
deep convection. Therefore, a comprehensive understanding of DCS dynamics requires the integration of
external datasets that capture precipitation, vertical cloud structure, and convective properties. By
combining geostationary IR observations with spaceborne radar measurements from TRMM-PR and GPM-
DPR, we can document not only the cold cloud shield but also the underlying deep convective processes
that govern the organization and life cycle of DCSs. However, while geostationary satellites provide
continuous monitoring of a given region, the low Earth orbit satellites TRMM and GPM offer only
instantaneous observations of specific regions as they pass over. A precise collocation procedure is then
required to integrate these datasets effectively.
In this study, we use Level-2 (L2) precipitation products from TRMM 2APR version 9 (V9) and GPM 2ADPR
version 07 (V07), covering 2012–2013 (TRMM) (Kummerow et al. 2000) and 2014–2020 (GPM)
(Skofronick-Jackson et al. 2017) within the 30°S–30°N region. Both datasets have identical spatial
resolutions, with a 245 km swath width and a 5.1 km horizontal resolution. The vertical detection range
extends from the surface to 20 km, with a 0.125 km vertical resolution across 176 levels. In order to ensure
continuity, harmonization of TRMM-PR and GPM Ku-band (13.6 GHz) calibration has been carried out,
reducing systematic differences and improving long-term precipitation estimates. These efforts have led
to a fully inter-calibrated and homogenized radar database, ensuring consistency across the 9-year (2012–
2020) record for the tropical belt (Stocker et al. (2018), Ji et al 2022). For our analysis, we focus on the
classification of precipitation type into stratiform, convective and other rain type, retrieved from the 2A23
algorithm (Awaka et al. 2009, Le and Chandrasekar 2013, Awaka et al., 2016, 2021; Chen et al., 2025).
The first step of the collocation procedure involves selecting, for a given geostationary platform, the IR
images that can be temporally collocated with a particular TRMM-PR or GPM-DPR orbit. Next, a spatial
collocation procedure assigns geostationary IR pixels to spaceborne radar pixels. Only geostationary pixels
identified as DCSs by the TOOCAN algorithm are retained for further analysis. Since the homogenized
GEOring dataset and TRMM-PR/GPM-DPR have different horizontal resolutions, we identify all GEO IR
pixels that fall within the TRMM-PR and GPM-DPR footprints to ensure a proper match. A temporal
collocation step is also necessary, considering the time-scanning characteristics of each geostationary
platform (Fiolleau and Roca 2020). To ensure the quality of our analyses, only geostationary pixels
collocated with radar observations within a 15-minute temporal window are retained. This ensures that
the matched IR and radar pixels remain as temporally aligned as possible.
DCSs identified by TOOCAN and sampled by TRMM and GPM radars are re-mapped onto the TRMM/GPM
geo-referential frame, ensuring that their morphological characteristics (e.g., size, lifetime, eccentricity…)
are analyzed within the same spatial and temporal reference as the radar data. This alignment allows for
a more consistent and accurate comparison of DCS properties across datasets. The brightness
temperatures from geostationary satellites are also averaged to match the TRMM/GPM radar footprint
resolution.
By combining these datasets, we establish a coherent framework that integrates convective cold cloud
shields identified from TOOCAN and GEOring IR data, with precipitation-related processes from radar
measurements. More than 900,000 DCSs have then been sampled by TRMM and GPM orbit at one or
multiple stages of their life cycle, but not at every time step, due to the intermittent nature of radar
overpasses. For our study, restricted to tropical ocean over a more restricted latitudinal band of 25°S-
25°N, only DCSs with at least 70% of their cloud shield sampled by TRMM or GPM radar orbit (≈225,000
DCSs) are retained (Fiolleau and Roca, 2013b). This selection criterion ensures sufficient spatial coverage
and enhances the statistical reliability of the dataset, allowing a more comprehensive and representative
characterization of DCS properties throughout their evolution.
Tropical cyclones that could bias the DCSs' properties were removed by discarding any DCS whose
centroid lay within 250 km of an IbTrACS storm track (Knapp et al., 2010), after collocating the tracks with
TOOCAN algorithm (Fiolleau & Roca, 2024). Following previous classification studies, we kept only cloud
systems with lifetimes longer than 5 h and a well-defined life cycle, corresponding to Class 2a in Roca et
al. (2017). Emphasis is placed on the convection that occurs on the first half of systems' life cycle (Elsaesser
et al. 2022), thus the analysis is further limited to snapshots between 10 % and 50 % of each DCS's
normalized life cycle. Over the study period this screening yielded ≈60 000 co-located DCSs in the TRMM-
PR/GPM-DPR radar dataset. Hereafter, this dataset will be called the TOOCAN-radar dataset. The
convective footprints of these collocated DCSs were delineated as illustrated in Fig. 1, which shows a
snapshot of a convective system detected by TOOCAN coincident with a PR/DPR overpass. Within the
radar swath, precipitation is classified as stratiform, convective, or non-precipitating following the
precipitation type from the 2A23 algorithm introduced before, and excluding the "other" rain type.
Convective echoes are highlighted in red in Fig. 1.

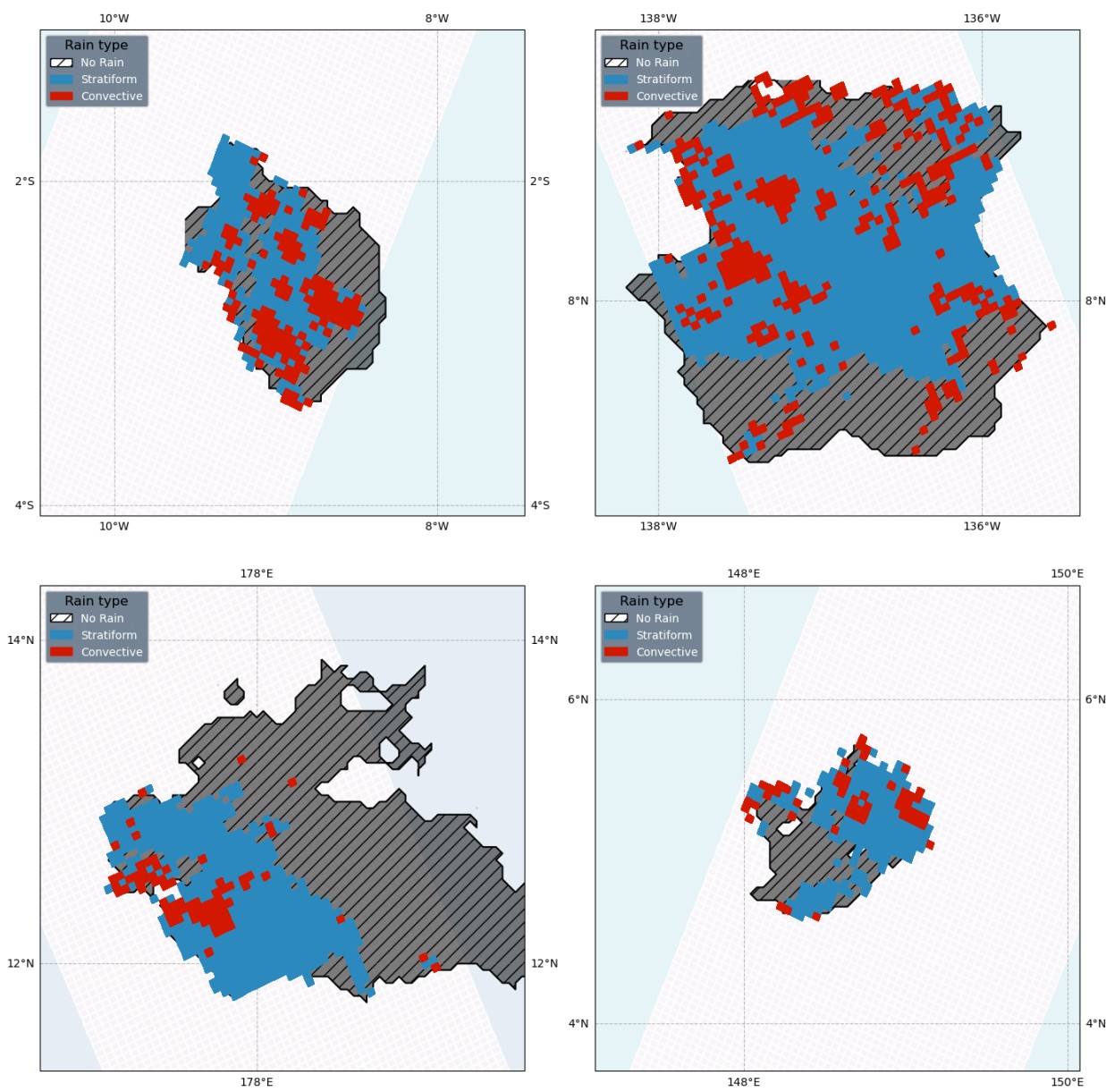

*Figure 1: Several examples from the TOOCAN-radar dataset. The TRMM/GPM swath along the track is shown in white, while the TOOCAN cloud shield at the time of collocation is represented in shaded-hatched grey. Different types of precipitation are displayed in color, with convective areas highlighted in red.*

## 2.2 Idealized model simulation

### The SAM model and the RCEMIP protocol

The cloud-resolving model System for Atmospheric Modeling (SAM) version 6.11.2 (Khairoutdinov & Randall 2003) is used. The simulation follows the RCEMIP protocol for CRM with a 300K SST (Wing et al., 2018) in a long channel configuration (6144x384km). The integration lasts 100 days and the last 25 days only are used and analyzed here to remove spin-up effects (Wing et al., 2020). The resolution is 3 km in both horizontal directions, and increases with height from a few tens of meters in the planetary boundary layer to 500 m in the mid and upper troposphere. All integrations are performed following the original

protocol with a specific output frequency of 30min instead of 1-3h and by using instantaneous fields rather
than hourly averages in order to permit the identification and tracking of convective systems in this
simulation.
To identify and track DCSs in the SAM simulation, the TOOCAN algorithm has been adapted to operate on
Outgoing Longwave Radiation (OLR) fields instead of traditional infrared brightness temperature (Tb)
imagery. While the core methodology of TOOCAN, based on spatio-temporal segmentation and iterative
thresholding, was preserved, the detection thresholds were defined to match the radiative properties of
the simulated convection. The initial convective seed detection was performed at an outgoing longwave
radiation (OLR) threshold of 73.90 W/m². The iterative region-growing process then expanded the
identified seeds using a 10-connected spatiotemporal operator, progressively including warmer
surrounding pixels up to a final threshold of 172.94 W/m², which corresponds to the upper boundary of
the cold cloud shield. This value is approximately equivalent to 235 K, following the Stefan–Boltzmann law
under the assumption of blackbody radiation. This adaptation enabled a consistent and physically
meaningful identification of DCSs in the high-temporal-resolution model OLR output.
Prior to applying the TOOCAN algorithm, a two-dimensional wrapping strategy was implemented to
ensure continuity in DCS tracking across the cyclic boundaries of the SAM domain. In the X-direction, the
domain was extended by duplicating the last 1500 km (corresponding to the final 500 grid points) and
appending it to the start of each OLR image. Similarly, the first 1500 km (first 500 grid points) were
duplicated and appended to the end. This symmetrical extension allows TOOCAN to detect convective
systems that traverse the zonal boundary without artificial segmentation. In the Y-direction, the entire
OLR image was duplicated and appended above and below the original field, effectively extending the
domain by 384 km in the meridional direction on each side. This ensures that vertically extended
convective systems near the Y-boundaries are also tracked without discontinuity. Following this wrapping,
any DCSs that were identified twice, once on each side of the duplicated X or Y boundaries, were carefully
merged or removed during post-processing to prevent double-counting in the final DCS database. This
approach ensures that the segmentation and tracking of DCSs are free from artefacts related to the
periodic geometry of the SAM simulation, enabling a reliable characterization of convective organization
across the full domain. Over the run period and 3 122 DCS have been identified in the simulation dataset.
## Convection mask
While convection is typically identified in cloud-resolving models using a simple diagnostic, such as vertical
velocity exceeding 1 m/s at a given altitude (Varble et al., 2014), a more sophisticated method is employed
here. Specifically, a classification approach is employed to distinguish between convective, stratiform, and
cirriform regions within the cold cloud shield. The method is based on a simplified implementation of the
physical threshold technique (Marinescu et al., 2016). It relies on a combination of surface precipitation,
profiles of vertical velocity and cloud top information to identify the cloud type using various thresholds
outlined in Table 1. A grid box is classified as cirriform if surface precipitation is below the stratiform
precipitation threshold and OLR is colder than a specific threshold, corresponding approximately to an
equivalent brightness temperature of 235K. A grid box is classified as convective if the surface
precipitation exceeds the prescribed convective precipitation threshold. Alternatively, it can be also
classified as convective if the absolute value of vertical velocity updraft (or downdraft) above (below) the
0°C isotherm exceed 5 m/s (3m/s). Grid boxes that do not meet the criteria for either convective or
cirriform classification are designated as stratiform.

| Convective Precipitation threshold (mm/h) | 0°C isotherm height (m) | Cloud top height (m) | stratiform Precipitation threshold (mm/h) | Cirriform OLR threshold (Wm-2) | Updraft threshold above the 0°C isotherm (m/s) | Downdrafts Threshold below the 0°C isotherm (m/s) |
|---|---|---|---|---|---|---|
| 10 | 3000 | 6000 | 0.04 | 172 | 5 | -3 |

*Table1 the physical threshold used for convective/stratiform/cirriform classification*
Hereafter, the TOOCAN-RCE dataset refers to the data introduced in this section. Examples from the RCE
dataset are shown in Figure 2, sharing a similar layout and color scheme than in Figure 1. Convective and
stratiform precipitation are rendered in red and blue, respectively. The convective echoes appear as
clustered patches whose boundaries are occasionally irregular.

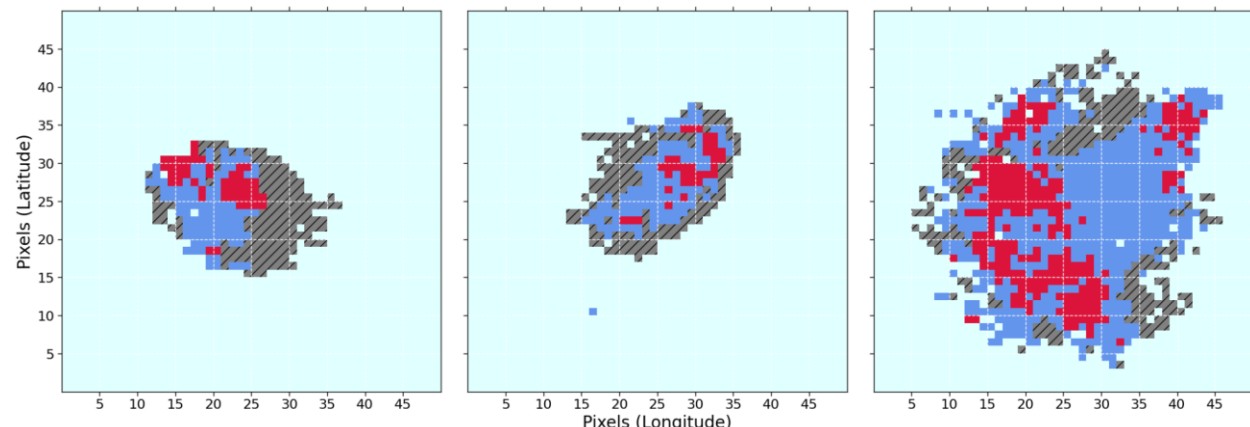


*Figure 2: Three illustrations of precipitation classification from the TOOCAN-RCE dataset, taken at a*
*selected time during the simulation within a 50x50 grid centered around the identified DCS. The TOOCAN*
*cloud shield is represented in shaded hatched grey. Different types of precipitation are displayed in color,*
*with blue indicating stratiform precipitation while convective areas are highlighted in red. The idealized*
*ocean is colored in light blue.*
**From convection mask to convective core identification and decomposition into elementary structures**
The concept of convective cores from radar observations has been popularized in the 1970's thanks to
the GATE field campaign that was dedicated to deep convective systems in the tropical Atlantic Ocean
(Houze & Betts, 1981). Surface based radar indeed revealed echo structures in cloud systems that
exhibited a strong coherency in space and time and intensity. The radar is sensitive to the hydrometeors
distribution that results from vertical movements so that there is a physical, although not direct, link
between the radar echoes and the underlying convective updrafts and downdrafts (Houze, 1997). Detailed
analysis of the radar echoes further indicate that the intensity of the reflected radar power is indeed
stronger at the center of the echo compared to its edge. This "core" structure mimics well its dynamical
equivalent where convective vertical velocity exhibits more intense value at their core since the updrafts
there are preserved from entraining less buoyant environmental air. The vertical extent of the cores can
reach up to the tropopause for the deepest ones and their spatial extension spans a wide range from 1 to
15 km for the deepest convective one (López, 1978). When extended to space borne radar, similar
features are observed. Based on earlier investigations, Houze et al. (2007) and Romatschake et al. (2010)
propose to define a convective core as a 4-connected (by shared edges) cluster of high intensity radar
pixels (5km). Such Houze-like cores, depending on the thresholds used, can span a wide range of spatial
scales, from 5 to 100 km. The physical and dynamical interpretation depends upon the size of the object
and some of its vertical characteristics and encompasses not only deep convection but a large spectrum
of convective activity and remains somehow qualitative (Chen et al., 2025; Houze et al., 2015). Figure 3
(left) presents a sample of radar measurements in which convective pixels are identified in red (see
Section 2, Data). Six convective cores have been labelled in Figure 3 (middle) using the Houze-like
clustering technique. These cores exhibit a range of sizes, from as small as 1 pixel to as large as 30 pixels—
approximately 150 km² for the green core.

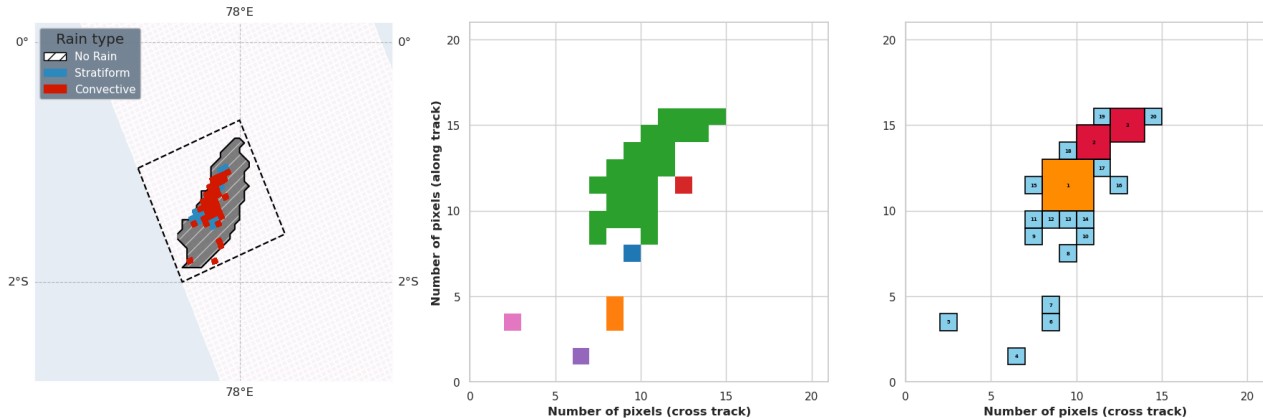


*Figure 3: Left: Same as figure 1 for another satellite scene, the black-dashed contour showing the
rectangular area that encompasses the cloud shield and defining the scene extent. Middle: basic 2D
segmentation of the convective field within the scene with shared edges, each entity is represented by a
color. Right: decomposition into basic square structures by decreasing size, each structure is represented
by a color and a number that is the edge length of the square.*

To move beyond this qualitative framework, we propose a finer-scale decomposition of the clustered echo
regions into elementary convective structures. This approach relies on the assumption that the spatially
continuous distribution of hydrometeors, and consequently of radar reflectivity, arises from aggregated
finer scale underlying dynamics. This approach can be viewed as an upscaled, space-borne version of the
analysis of precipitation core and updrafts/downdrafts joint occurrence analysis from high-resolution
ground-based observations (e.g., Moroda et al., 2021; Lamer et al., 2023). In this perspective, we assume
that the large cluster of continuous echoes (colored in green in Figure 3, middle) is composed of smaller
coherent and compact convective features akin to a circular bulk updraft of varying diameter that we
approximate using size varying squares. We hence add a morphological constraint to the classical core
segmentation.
The decomposition technique is further detailed in Annex A. An example of the decomposition is shown
in Figure 3 (right) where the 30-pixel central core, as identified by the Houze-like methodology, is now
broken down into 13 single pixel elements, two 2x2 pixels elements and one 3x3 pixels element. In this
example, the maximum size of square elementary structure (MaxSquareSize) that fits in the convective
mask is 3 (Figure 3 middle).  Our decomposition technique, as well as the Houze-like clustering technique
have been applied on the TOOCAN-radar and TOOCAN-RCE datasets. The core size distribution confirms
the difference between Houze-like clusters and the results from the decomposition into elementary
square structures (Figure 4). For the TOOCAN-radar dataset, the Houze-like cores are log-linearly
distributed with a maximum size up to 17x17 pixels equivalent (roughly 7200 km²) while the present
decomposition prevents clusters larger than 9x9 pixels, roughly 2000km2 (Figure 4, left).

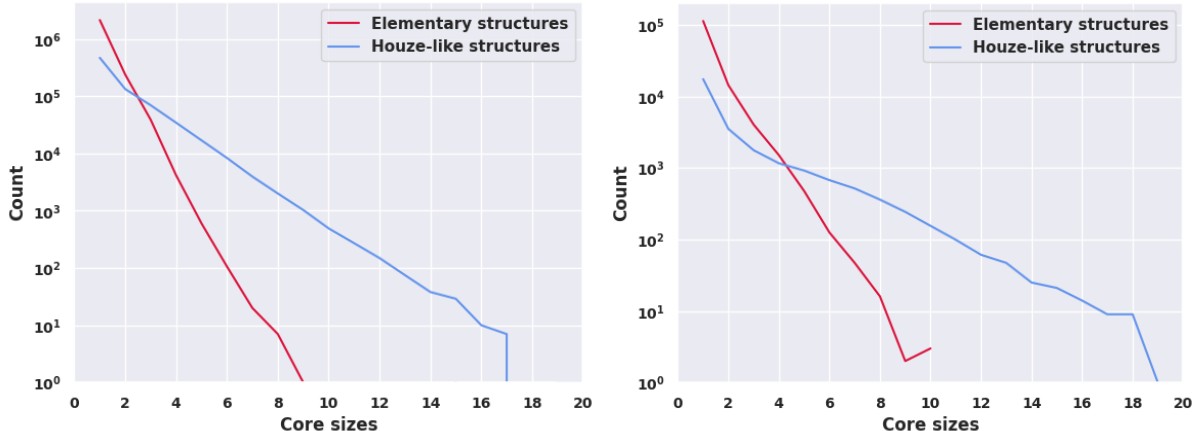


*Figure 4: Size distribution of the convective cores Left: TOOCAN-radar dataset. Right: TOOCAN-RCE*
*dataset. The Houze-like detection is shown in blue and corresponds to the rounded integer value of the*
*square root of the cores' area, while our elementary decomposition is shown in red and corresponds to*
*the size of the square cores.*

Similarly, to the radar observations, km-scale model simulations can be analyzed in terms of elementary
structures. While resolving the deep convective updrafts/downdrafts dynamics requires hectometric
resolution (Bryan et al., 2003), km-scale models have been shown to produce reasonable vertical
velocities and precipitation structures (Kukulies et al., 2024). At these spatial resolutions, though, the
updrafts/downdrafts correspond more to a bulk plume associated with a coherent hydrometeors loading
across a few grid points (Varble et al., 2014) , similar to what is observed in the radar data. As exemplified
in the next section, our technique successfully decomposes the convective mask from the simulations (see
section Data above) into elementary structures like the one illustrated in Figure 3 (middle). Statistically,
differences similar to the radar case hold for the TOOCAN-RCE dataset (Figure 4, right) with elementary
structures reaching maximum size up to 10x10 grid points, roughly 900km2.
In both datasets, the detection of the convective cores thanks to the decomposition in elementary
structures prevents the building of very large patches of convection that are delicate to interpret
physically. While a thorough exploration of the better dynamical and physical consistency of the present
core identification compared to a simple 4-connectivity segmentation is deferred to future work, Figure 5
shows a comparison of the mean vertical profile of reflectivity for the present decomposition and that of
Houze-like cores, using 8 years of collocated radar measurements to allow for robust statistics. The
present decomposition reflectivity shows a pattern similar for all cores sizes. It is composed of an almost
constant distribution from surface up to the freezing level (≈5km) and then a slowly decreasing reflectivity
up to the maximum altitude of deep convective reflectivity typical of deep convective structures (e.g.,
Zipser et al. 1994). This pattern scales well with the size of the core, showing well discriminated profiles
as the size increases. Houze-like cores, in contrast, show overall weaker mean reflectivities. Also, the
Houze-like profiles exhibit much less variation with the size of the core than the present decomposition.
The stronger reflectivity of the present decomposition, for a given size, suggests more homogenous
profiles being aggregated together resulting in the enhanced mean profile compared to Houze-like cores.
The two decompositions also differ in how well they resolve size-dependent physics. In the
elementary-structure case, the altitude of the 30 dBZ isosurface rises steadily from ≈ 4 km for 2×2 cores
to ≈ 9 km for 8×8 cores, , indicating that larger structures are associated with stronger and deeper
convection (Fig. 5, left). Houze-like cores smaller than 6×6 never reach 30 dBZ, and larger ones show no
clear trend (Fig. 5, right). Similar comparisons using the simulation's vertical velocities confirms the lack
of coherency of the large patches from the Houze-like segmentation (not shown).

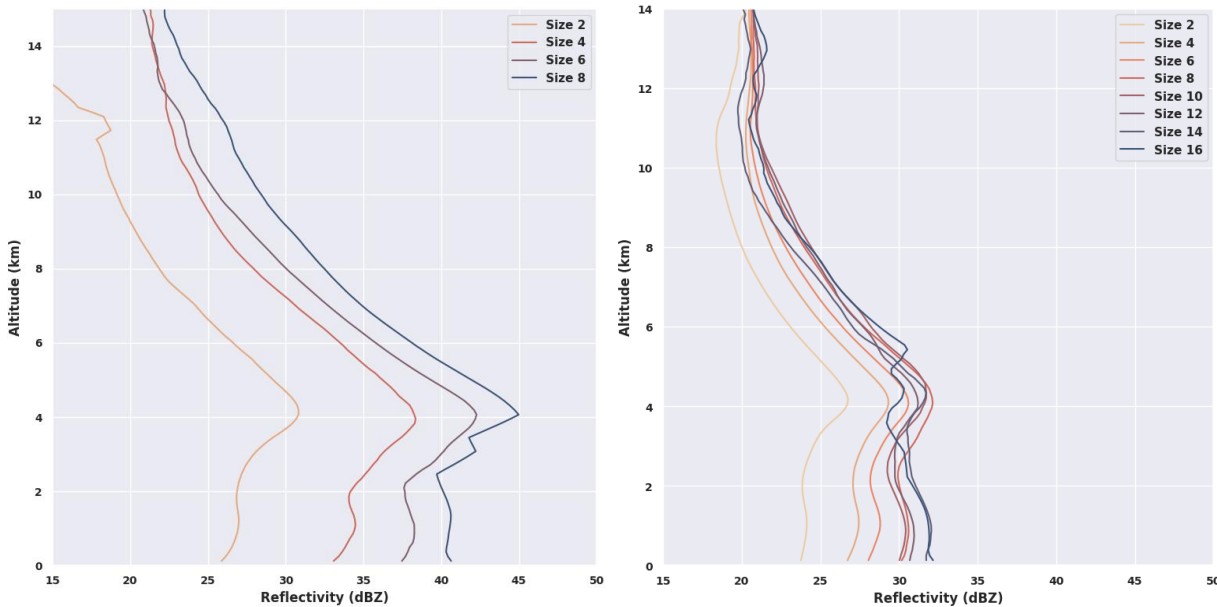


*Figure 5: Reflectivity profiles (composites using 8 years of collocated radar measurements) of convective*
*cores for Left: our elementary decomposition, Right: Houze-like shared edges decomposition, with*
*different core sizes shown in color.*
We acknowledge that this decomposition relies on strong assumptions about the underlying dynamics
behind the radar echoes or model convective mask clusters. Nevertheless, the present effort is one step
toward improving the consistency between the core identification and the underlying dynamical object.
As demonstrated in the next section, this approach significantly enhances the characterization of the
organization of convection within the cloud shield of DCSs.

# 3 The method

## 3.1 Rationale

The aim is to classify a scene based on the spatial arrangement of the convective cores within the convective system's cloud shield. The cloud shields span a wide diversity of scales and shape (Roca et al. 2017) and the density of convection across the shield also exhibit large variations (e.g., Elsaesser et al., 2022). As a consequence, unlike regular gridded data of fixed size, the method requires defining a selected area that corresponds to the cloud shield dimensions. To address this requirement, we qualify the spatial arrangement of convection in the cloud shield by comparing the actual scene with an ensemble of generated scenes for which the spatial arrangement of the cores is randomly distributed in the shield. The generated scenes have the same characteristics as the actual scene in terms of convective fraction, distribution of the size of the cores, and differ only by the spatial arrangement of the cores. This measure of a deviation from a random state is inspired from i) the work of Koren et al. 2024 where they applied their methodology to a 2D cloud mask within a selected area, ii) the organization irregularity index (OII) from Biagioli et Tompkins 2023 that is a function of the distribution's departure from randomness across the full spectrum of spatial scales. This stochastic approach allows for the estimation of the probability that a given arrangement of convective cores could arise purely by chance. This metric, together with the characteristics of the scene (see dedicated part in section 3.2) forms a small set of key parameters (Janssens et al. 2021) that are used to characterize the organization of convection using an unsupervised simple classification procedure. In the following, the algorithm and its domain of applicability are presented in detail.

## 3.2 The algorithm

*Figure 6: Overview of the algorithm to compute the 4 key parameters of a scene.*

Figure 6 shows the structure of the algorithm used to derive the 4 key parameters corresponding to a given scene. The following sections describe each step of the procedure and details the computations that are represented by the colored arrows in Figure 6.

## Scene characterization

### *Definition*

A scene is defined as the rectangle that most closely frames the outline of the cloud shield identified by the TOOCAN algorithm, as illustrated by the black contour in Figure 3 (left). For the TOOCAN-Radar dataset, this rectangle must be oriented in the direction of the satellite's orbit within the TRMM/GPM swath geometry. It is not possible to extend beyond this geometry because radar information is missing there. When the data is available, a margin of 1 pixel is also used at the edges. Assuming the individual pixel area is similar across the grid, the total surface of the scene is expressed in pixels. We introduce four parameters to quantify convective organization. The first two parameters describe the amount of convection within the scene, using both absolute and relative metrics.

### *Convective area A and convective fraction F*

The convective area A corresponds to the number of convective pixels in the scene. The convective fraction F is the proportion of those pixels regarding the area of the scene and is expressed in percent (upper-left box of figure 6).

### *Characteristic length of convection arrangement in the scene*

A characteristic length scale is computed to summarize the macroscopic spatial coherence of convection within the scene. It is defined as the maximum distance between any two pixels in the spectrally transformed version of the scene. This metric will later be used to compare the observed scene with the corresponding generated one.

The length L is hence obtained by applying various filters on the scene (upper line workflow in Figure 6). The 2D autocorrelation spectral power field of the scene is computed and normalized (yellow arrow in Figure 6). The 2D autocorrelation power is illustrated in Figure 7 (and Figure S1) for two typical scenes selected from the satellite dataset. The compactness of the scene shown in the top panel, is expressed by a strongly marked central area in the 2D power map, while the second scene, in the bottom panel displays a more spread feature indicative of lower spatial autocorrelation. Although two-dimensional autocorrelation and Fourier analysis are mathematically related, we favor autocorrelation because it is less sensitive to the noise and spurious features that commonly appear in convective fields (Figures 1 and 2).

The first filter, applied on to the normalized spectral power field, is a simple threshold of 10% to delineate the region of interest within the scene. Only the central shape is considered hereafter (Figure 7, bottom-right). To handle the rare cases for which this contour would exhibit complex, twisted, or distorted shapes, the smallest convex contour that encloses it, is identified (blue contour in Figure 7). The characteristic length L is finally obtained as the maximum distance between any two pixels inside the convex contour (orange arrow in Figure 6).

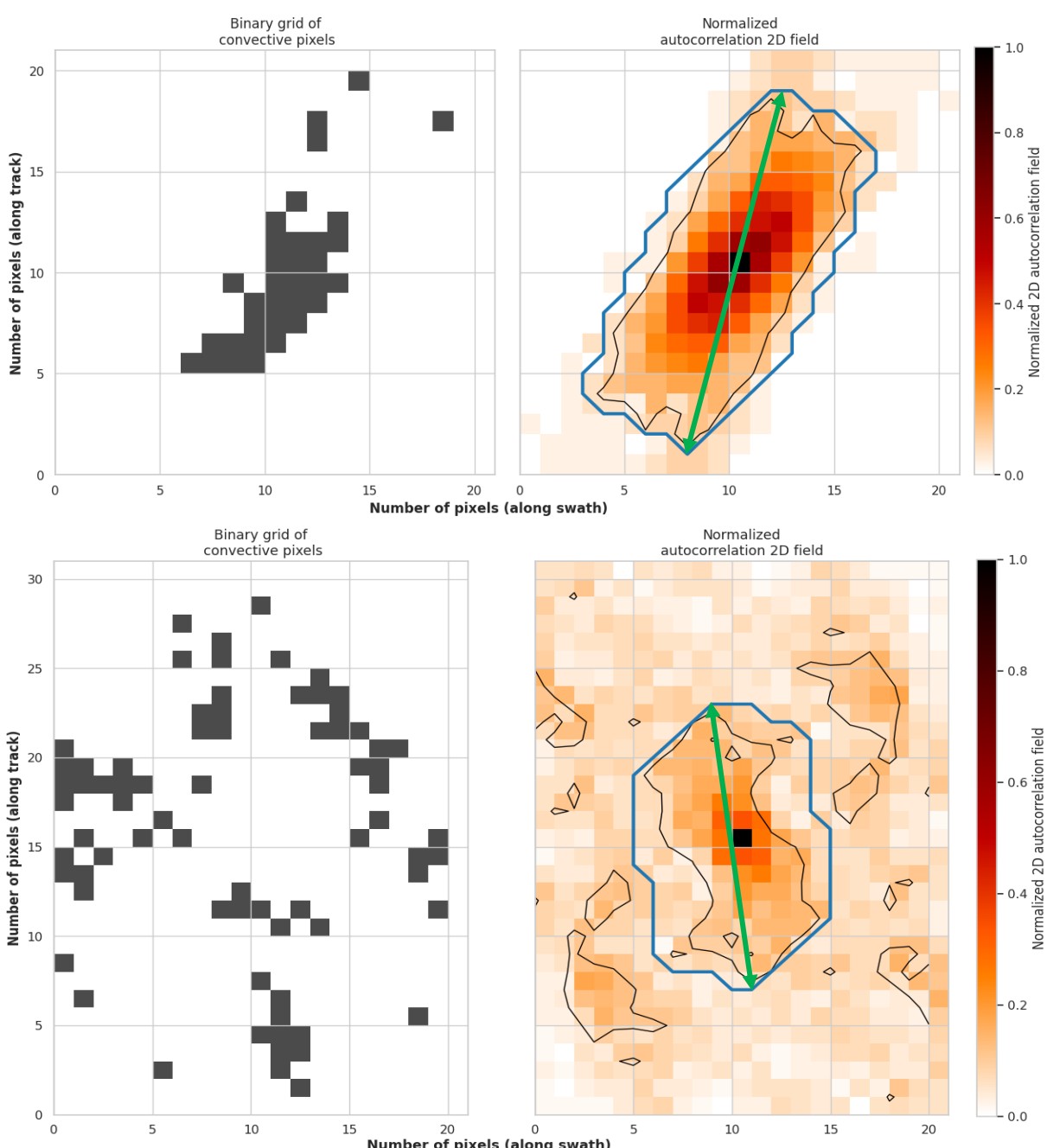

406

*Figure 7: Two selected scenes within the TOOCAN-radar dataset. Top: More compact convective field, Bottom: More spread convective field. Left: 2D binary scene (black pixels represent convective areas). Right: 2D autocorrelation normalized field, black contour shows the 10% contour and blue contour represents the convex contour that encompasses that 10% contour. The characteristic length L is the maximum distance between 2 points within the blue contour, represented with the green arrow, here top: L≈18.44, bottom: L≈14.94 (pixel unit).*

This isocontour, centered on the peak of the power map, delineates the spatial extent where spatial correlation remains significant. It marks the threshold beyond which structural repetition becomes

negligible, thereby providing a measure of the decorrelation scale, e.g., the characteristic distance at which recurring motifs lose coherence. The contour's shape reflects anisotropy. A near-circular form indicates isotropy with uniform repetition in all directions, while elongated or asymmetrical contours reveal directional dependence. Its size quantifies how spatially spread the pattern is within the image and results from a balance between spatial arrangement and overall convective fraction F within the grid. Other examples of this process using the 2D autocorrelation fields are shown in Figure S1 (supplementary materials).

The extraction of L is performed in the 2D autocorrelation space, rather than directly from the raw convective field. As a result, L encodes both the perception of the spatial extent of the main structures of the raw field and the condensed structural information regarding the spatial arrangement of these structures captured by the autocorrelation. However, the signal-to-noise ratio and the area of validity of this 2D field must be carefully assessed to ensure a reliable interpretation of the L values as defined here. Sensitivity to the arbitrarily selected threshold and limits of applicability are further discussed in Section 3.2.4 and in Annex B.

C and A are two of the 4 key parameters to characterize a scene and are independent from the relative spatial arrangement of the cores. They contain information on the overall filling of convective structures within the cloud shield, while the parameter L contains information on the organization for given F and A. The L scale will be ultimately compared to that derived from stochastic modelling, including scene generation, to calculate a percentile-based measure of deviation from randomness.

## Scenes generation

In this section, the stochastic approach is described, along with the definition of the fourth key parameter, P, that represents a percentile-based measure of the scene's spatial arrangement to deviate from a random distribution (lower line workflow in Figure 6). The decomposition in elementary structure is first performed (See Section 2 and Annex A, green arrow in Figure 6) and used in a Monte-Carlo approach for generating a reference distribution under spatial randomness (blue arrow in Figure 6).

## Monte-Carlo approach

The geometric properties of the scene, such as its size, convective fraction F, and the cores from the decomposition into elementary structures, are used to generate synthetic scenes, in which the segmented cores are randomly distributed in space following a uniform distribution. This approach is conceptually similar to that of Haerter et al. (2019), who also used a uniform law to position the centers of "cold pools" in fixed grid sizes. Importantly, the original convective fraction F and convective area A of the convective cores are preserved in each generated synthetic scene, only their spatial arrangement differs. Figure 8 illustrates the procedure: from the scene (a), the elementary decomposition is performed (b) and two generations of the same core's partition are randomly displayed (c, d). The first realization results in low L values, while the second, where the largest cores are more spatially clustered, yields higher L values. This demonstrates that in some cases, random distribution can produce spatial arrangements that are comparable in terms of L values, to that of the original scene.

An ensemble of 500 synthetic scenes, for satisfying statistical robustness (see Annex B), is generated and for each of them, the characteristic scale L is computed as described previously. Thus, a synthetic distribution of random surrogates of the original scene has been created and the L distribution of these

random scenes is assessed (Figure 9, purple arrow in Figure 6). This distribution serves as a reference for evaluating the degree of spatial arrangement in the identified scene relative to the generated ensemble.

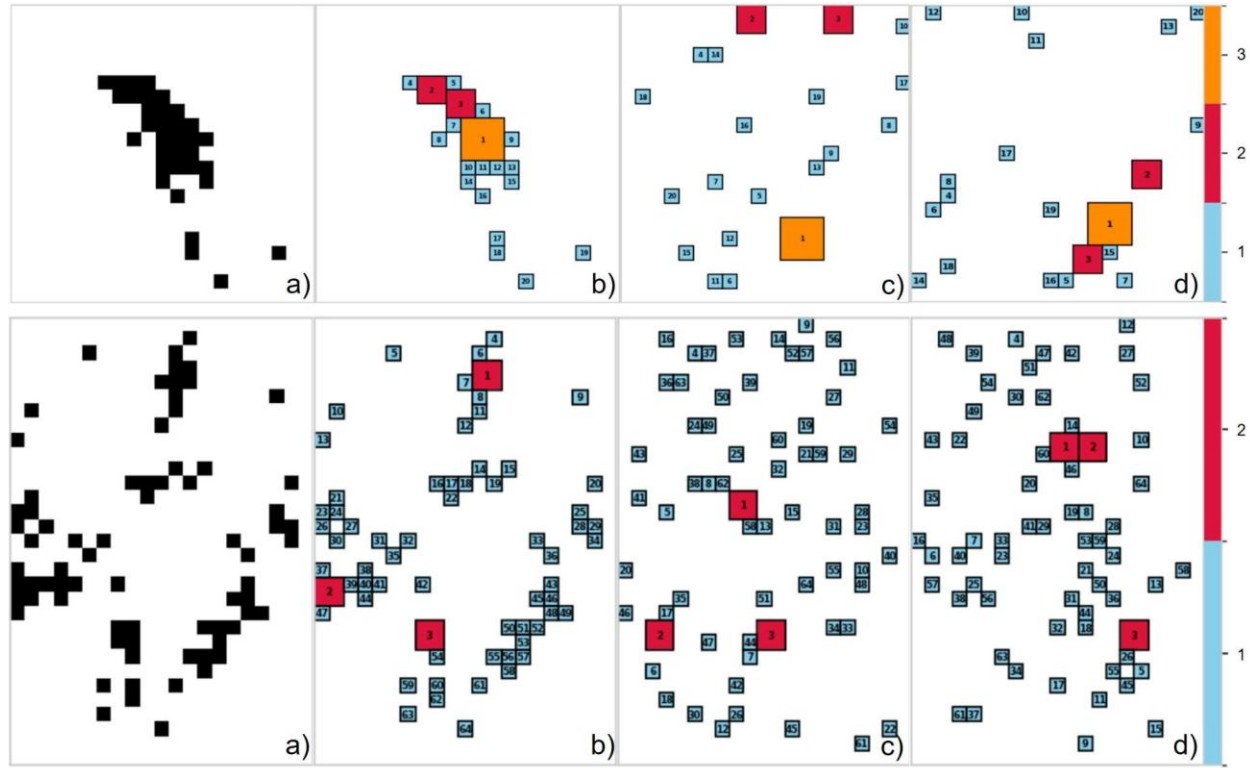

*Figure 8: a) 2D binary scenes as shown in Figure 7. b) elementary decomposition into square clusters. (c,d) : Generated scenes with the same number of square clusters, randomly and uniformly distributed, the c/d panels are respectively a scattered/clustered random positioning that produces lower/higher L value.*

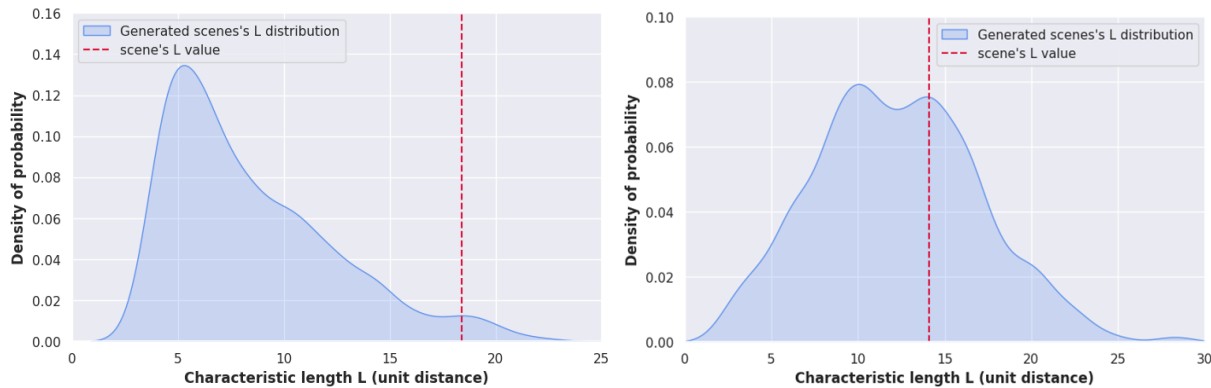

*Figure 9: Left: (respectively right) Empirical probability density function (estimated via KDE) of the generated distribution of characteristic length L for the scenes in Figure 7, top (resp. bottom). The scene's L value is represented with a vertical dotted red line.*

*Computation of P*

The comparison of the identified scene to the ensemble of generated length scales L is performed by computing a percentile P, which quantifies the deviation of the real scene's spatial arrangement from a random arrangement (Figure 9, Red arrow in Figure 6). This measure is estimated by computing the percentile of the scene's L within the generated distribution. For instance, a value of P = 0.1 indicates that 450 out of the 500 generated random scenes have an L value greater than that of the real scene. Conversely, a value close to 1 means that almost no randomly generated scenes produce a characteristic length L greater or equal than the one from the scene.  This implies that such a specific spatial arrangement (or its equivalent) almost never occurs in the randomly generated scenes. The combination of P with the 3 other parameters provides a fully comprehensive characterization of the complexity of the spatial arrangement for a given scene (Figure 6).

## Limits of applicability

In some cases, the quantification of spatial arrangement of convection is an ill-posed problem and provides little if any insights into the underlying physics. This happens when the scene has obvious limitations in size or convective fraction. It can also arise from sensitivity of the stochastic model to some of these characteristics of the scene. Indeed, the generation of the ensemble of scenes given a convective fraction (F) and convective area (A) is constrained by a few parameters of the stochastic model, namely, the number of realizations, the threshold used for contour identification during the computation as well as the maximum size of the elementary structure (MaxSquareSize). The sensitivity of F to the two former parameters are discussed in Annex B while this section focuses here exclusively on the sensitivity to MaxSquareSize, as it is the primary influencing factor.

In the case of the narrow satellite observations of the radar, the estimation of the shield area (and, by consequence, the F parameter) using an along track aligned rectangle yields a biased estimation of this area. Nevertheless, this 'algorithmic' variable correlates very strongly with the true area and does not overall impact our methodology. But, in the case of elongated systems (eccentricity < 0.3) with a specific alignment with respect to the along track, in the range of 30° to 45°, this effect could be significant. We underscore the need to use our technique with caution for these infrequent cases: combination of cloud shield morphology and relative observations configuration.

Idealized simulations were carried out to estimate the variations in the distribution of length as a function of the F parameter and varying MaxSquareSize values. Using a fixed 50x50 pixels grid, the distribution of length generated for a given F value and a selected MaxSquareSize are analyzed to determine whether the percentile of the L value of the scene could be accurately assessed. The parameter MaxSquareSize has a significant impact on shaping the L distribution, as shown in Figure 10. The mean of L is an increasing function of F, as expected. However, the growth is non-linear. Specifically, it becomes challenging to compute a meaningful percentile when L values are confined near 0 (e.g., for low F and MaxSquareSize = 1, as shown in Figure 10, left) or when L values saturate at the grid's diagonal length, $(49^2+49^2)^{0.5} \approx 69.3$, which occurs for high F values and MaxSquareSize ≥ 5 (Figure 10, right).

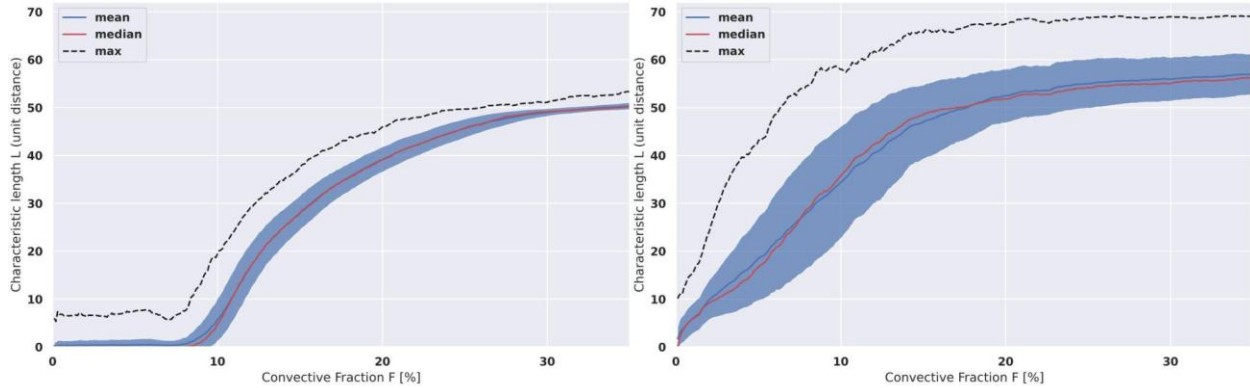

503

*Figure 10: Characteristic length L plotted against convective fraction F. 500 generation of 50x50 grids for Left: only square clusters of size 1 (MaxSquareSize = 1), Right: same but for square clusters partitioned with MaxSquareSize <= 5. The red line indicates the median of the distribution, the dark blue line shows the mean and the dotted-black line represents the maximum value of L within the distribution. The shaded blue area represents +/- 1σ. The x-axis resolution is 0.1%, with a 1.5% rolling mean applied for readability.*

Based on these diagnostics we limit the applicability of the algorithm to the scene whose convective fraction lies between two empirical bounds: Minimum convective fraction $F_{min}$(MaxSquareSize) = 8% for MaxSquareSize ≤ 1 and Maximum convective fraction $F_{max}$(MaxSquareSize) = 25% for MaxSquareSize ≥ 5. Scenes that do not satisfy these thresholds are discarded, guaranteeing that the subsequent characterisation remains physically meaningful and robust.

## Data processing

A minimum scene size criterion is also used to filter out scenes with less than 8×8 grid size and less than 5 convective pixels, as characterizing scenes below these thresholds provides limited insight into the spatial arrangement of convective structures. Therefore, very small scenes or scenes containing sparse convective areas are not characterized. After applying those filtering, roughly 20% of the TOOCAN-radar dataset is filtered out. This number is down to 5% for the TOOCAN-RCE dataset. The final datasets then consist of 54 132 scenes for the TOOCAN-radar dataset and 2 941 scenes for the TOOCAN-RCE dataset. For each of these scenes, the four key parameters are computed to characterize their spatial arrangement, which serves as the basis for the classification. Table 1 summarizes the baseline configuration used to perform these computations prior to the final classification step, while duration and maximal size distributions of DCSs selected in both datasets are shown in Figure 11. In both datasets, the population of DCS under consideration spans a wide range of morphology for short to long-lived and from small to very large systems confirming a good diversity of scenes for the analysis.


| Threshold for convex hull identification | 10% |
|---|---|
| Number of Monte-Carlo realizations M | 500 |

| Minimal grid size | 8x8 |
|---|---|
| Minimal total convective area $S_{min}$ | 6 pixels |
| Minimum convective fraction $F_{min}$(MaxSquareSize) | 8% when MaxSquareSize ≤ 1 |
| Maximum convective fraction $F_{max}$(MaxSquareSize) | 25% when MaxSquareSize ≥ 5 |

*Table 2: baseline configuration.*

## 3.3 Results

### Diversity of the scenes

More than 54000 and 2900 scenes are available for the TOOCAN-radar and TOOCAN-RCE datasets
respectively. The morphology of the cloud shield of the DCS associated to these scenes is summarized in
Figure 11. For both the TOOCAN-radar and TOOCAN-RCE datasets, the duration and maximum area (in
km²) span a wide range of values consistent with the unfiltered distribution (not shown) suggesting that
the population of scenes under analysis is representative of the DCS distribution.

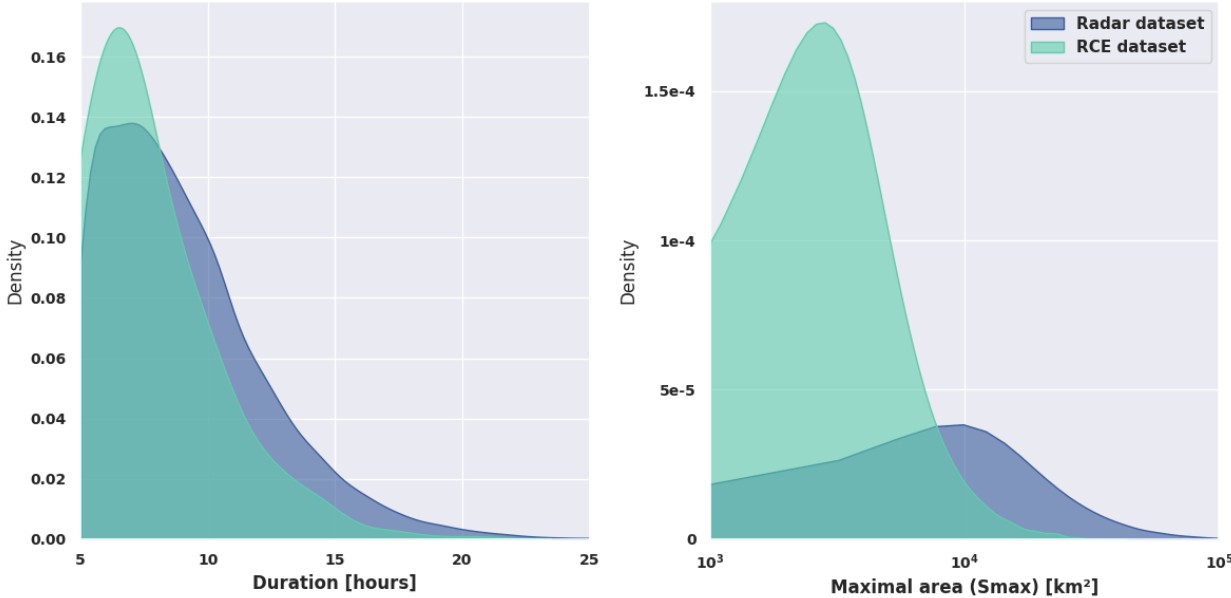

*Figure 11: Empirical probability density function (estimated via KDE) of DCS properties from the scenes.*
*Left: Duration in hour, Right: Maximum area in km² for the TOOCAN-radar dataset (dark blue) and the*
*TOOCAN-RCE dataset (light blue).*
The four key parameters' distribution are shown in Figure 12, for both datasets. While the DCS and their
internal organization from idealized simulations cannot be compared one-to-one with the TOOCAN-radar
dataset, the similar distribution of F, A, L and P suggests a very good behavior of the SAM model in
simulating DCS. This point has already been noted either in global CRM configuration (Feng et al., 2025;
Abramian et al., 2025) as well as in RCE configuration (Roca et al.,2024).
Figure 12 indicates that the convective fraction, size of convection area and characteristics lengths of the
scene of the datasets span a large range of values, with slightly skewed distributions. These distributions
are indicative of a wide diversity of spatial arrangement of convection in the scenes, both for the radar
and the simulation. On the other hand, the P distribution is very much skewed to the highest values
(around 1) with a very long tail (Figure 12, d). This indicates that most of the scenes exhibit a characteristic
length that cannot be easily reproduced by random generation. The fact that the general features of these
distributions are found in both the TOOCAN-radar and the TOOCAN-RCE datasets further suggests that
the use of the four parameters to characterize the scenes allows to address the diversity of the convective
organization, independently of the marker used to identify convection, from the hydrometeors in the case
of the radar and from the vertical velocity for the simulation.

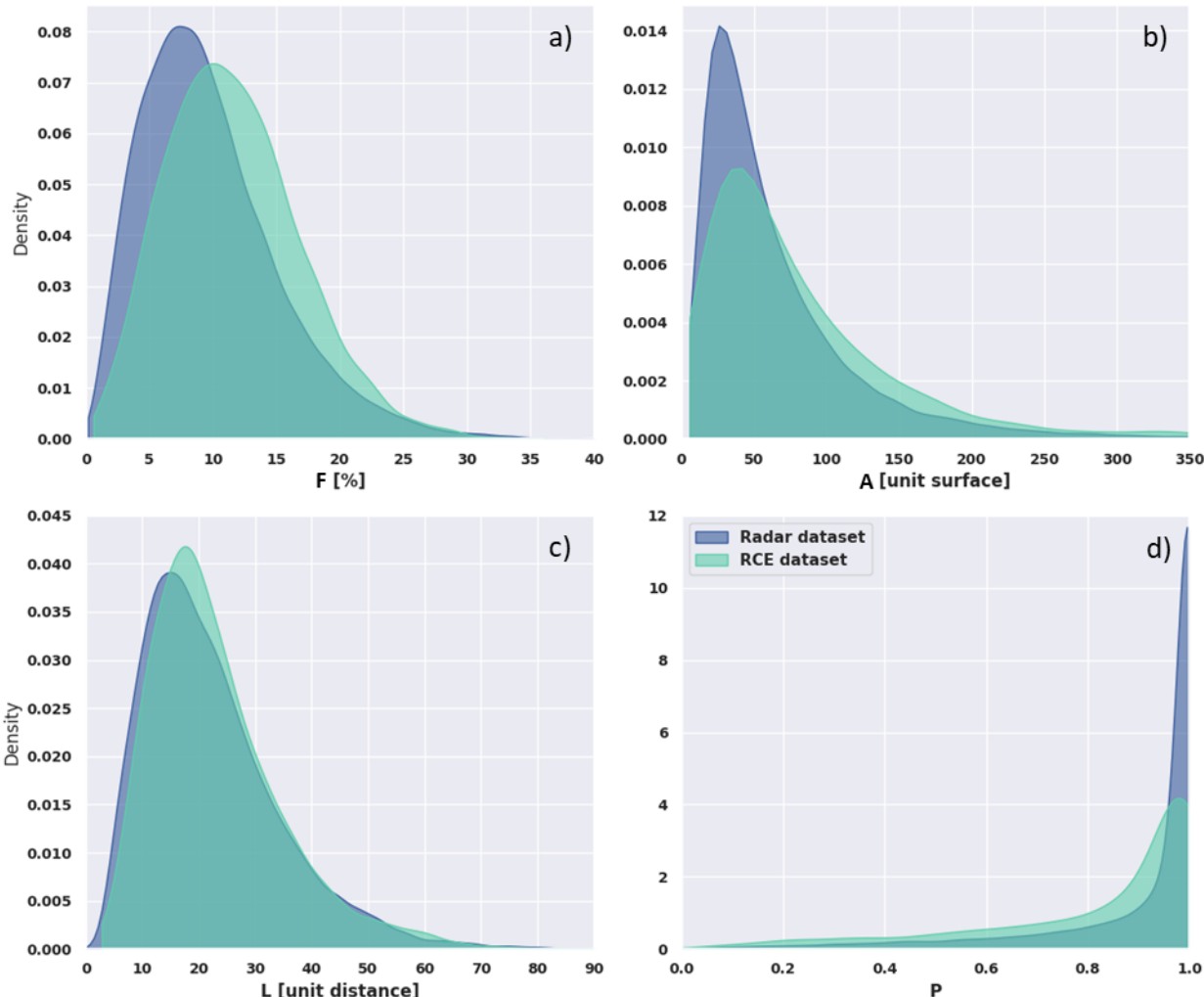


*Figure 12: Empirical probability density function (estimated via KDE) of the 4 key parameters within the*
*TOOCAN-radar dataset (dark blue), the TOOCAN-RCE dataset (light blue). From Top to Bottom, Left to*
*Right: convective fraction (F), convective area (A), characteristic length (L) and percentile (P).*

## Organization of convection

The organization of convection can now be related to the multivariate coherency of our 4 parameters to
characterize a scene. This 4-dimensional space is analyzed thanks to a simple unsupervised classification

technique (K-mean). Four classes are hence automatically built for each of the datasets. Classical clustering metrics (a trade-off between elbow and silhouette scores) show optimal results when separating the scenes into four classes. The results indeed reveal well separated classes, showing as separated pairs of distributions for each parameter (Figure 13). As shown in Figure 13 d and h, three of the classes are associated with the high values of P and one class stands out as possibly randomly arranged scenes, corresponding to the small occurrences of the long tail of the distribution (less than 20% see Figure S2).

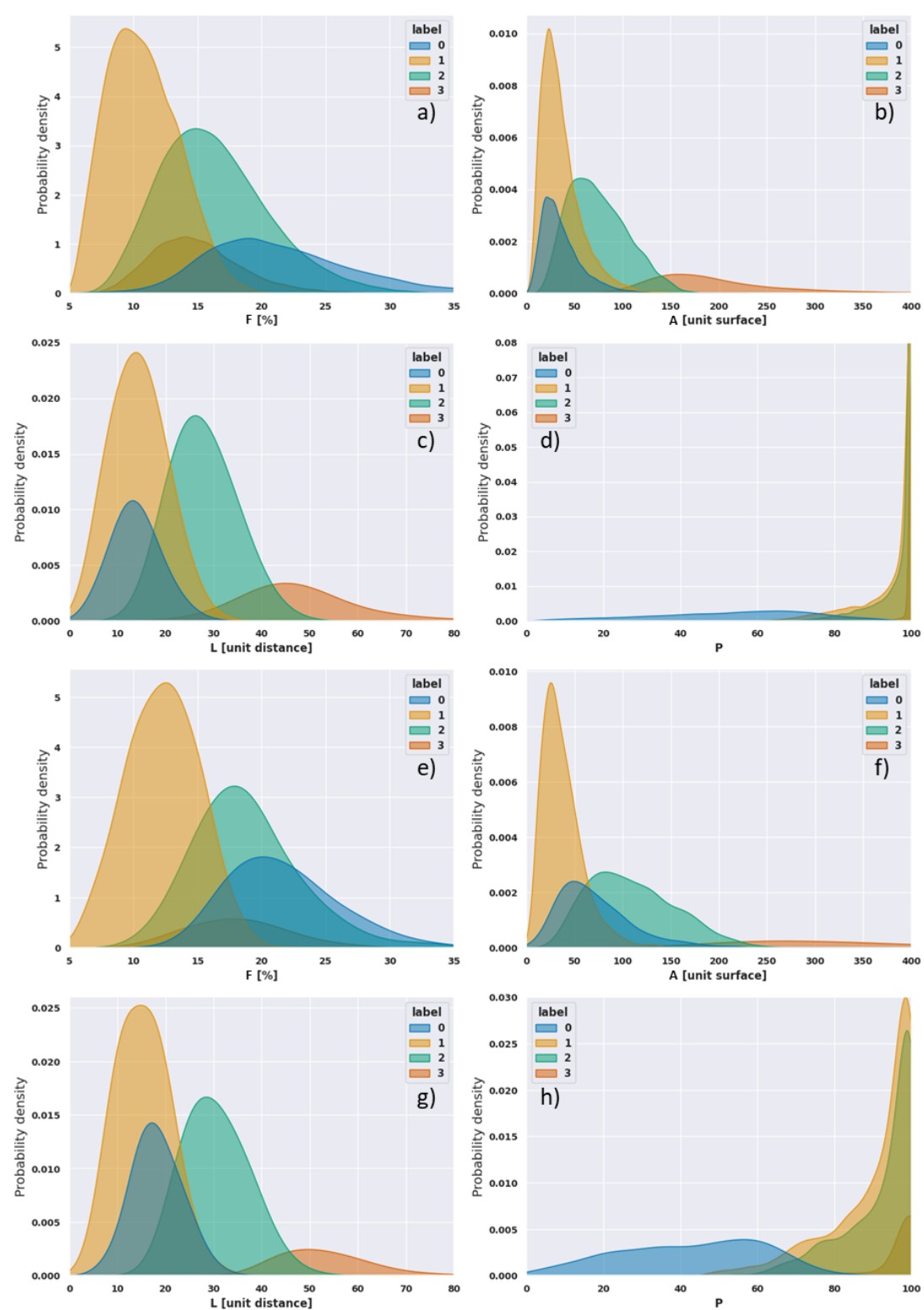

570

*Figure 13: Empirical probability density function (estimated via KDE) for each class. Top: Convective scenes from the TOOCAN-radar dataset, Bottom: Convective scenes from the TOOCAN-RCE dataset. a, e) convective fraction F, b, f) Total convective area A (in pixels), c, g) Characteristic length L (in distance unit), d, h) Percentile P.*

An increase in P, together and secondary with the variables L and A, corresponds to a higher level of
organization. Accordingly, the classes are ordered from class 0, representing the least organized state, to
class 3, representing the most structured. More specifically, class 0 (in blue) represents convective
organizations that are the least clustered (characterized by relatively small L and P), even when F is high.
In contrast, the class 3 (in dark orange) exhibits the most structured convective areas (high L, A, and P),
despite relatively small F values. This indicates that F alone is not sufficient to fully discriminate between
organization types, as the spatial arrangement strongly influences the other three key parameters. Highly
organized scenes (very unlikely to be randomly spatially distributed with high L values) do not have high
F values, as the dominant factor is the arrangement of the structures inside the cloud shield. In contrast,
a clear relationship between L and A is observed, with both parameters varying consistently across the
four classes, as expected (Figure 13). The primary distinction between the two intermediate classes lies
less in their level of organization (both share similar P distributions) and more in the extent of the
convective areas. Class 1 (in yellow) corresponds to scenes with relatively small convective areas (low F
and A), leading to smaller L values compared to the class 2 (in green), which represents scenes with larger
convective areas. The Class 0 and 3 represent the extreme opposites in terms of organization, and they
are clearly separated in the A, L, and P spaces. The Class 0 scenes differ from the Class 1 primarily in F and
P, while the Class 2 and 3 scenes are mainly distinguished by A and L. The least distinct separation occurs
between class 1 and 2 scenes, as they only partially differ in F, A, and L. The Class 1 scenes systematically
exhibit lower values than Class 2 for these parameters, suggesting that these two organizational types are
closely related. While Class 1 represents a less pronounced version of Class 2 in terms of convective area
within the cloud shield, their spatial arrangement remains comparable, as indicated by their comparable
P distributions. Finally, beyond a certain threshold of A (depending on the dataset), the scene can only be
of Class 3 (Figure 13). This means that the most structured type of organization always occurs once there
is enough convective area within the scene. This also indicates that for the majority of A values, knowing
the values of the two parameters F and A is not enough to properly separate different types of
organization. The overall classification is highly consistent across the two datasets, showing similar class
distributions within the four-dimensional space and comparable general behavior. Again, despites
differences in the marker used to identify convection, this suggests that our method gives rise to well
separated classes to handle the diversity of scenes found in both datasets.
Figure 14 presents archetypal scenes from the TOOCAN-radar dataset for each class, selected by
identifying the scene closest to the class centroid in the four-dimensional variable space. As expected,
Class 0 displays a popcorn-like pattern characteristic of weakly sheared environments (e.g., Anber et al.,
2014). Class 1 corresponds to a small system with a compact convective zone, while Class 2 exhibits a
similar structure but on a larger spatial scale. Class 3 shows well-organized convection, exemplified here
by a linearly structured convective system (e.g., Houze, 2004). Based on this classification, we propose the
following class labels: Likely Random (label 0; blue), Clustered and Small (label 1; yellow), Clustered and
Large (label 2; green), and Very Structured (label 3; dark orange). Their abbreviated names are LR, CS, CL,
and VS, respectively.
The strong discriminating power, illustrated with this visual example (other examples for the TOOCAN-
radar dataset can be found in Figure S3 and in Figure S4 for the simulation), of our method revealed
here thanks to a basic unsupervised classification technique is the results of the selection of our 4 key
parameters to characterize the scenes. These four features indeed contain enough information to
discriminate between random and organized spatial arrangement. In particular the detection of cores
thanks to the decomposition into elementary structures helps the quantification of the degree of
organization compared to a randomly distributed scenario.

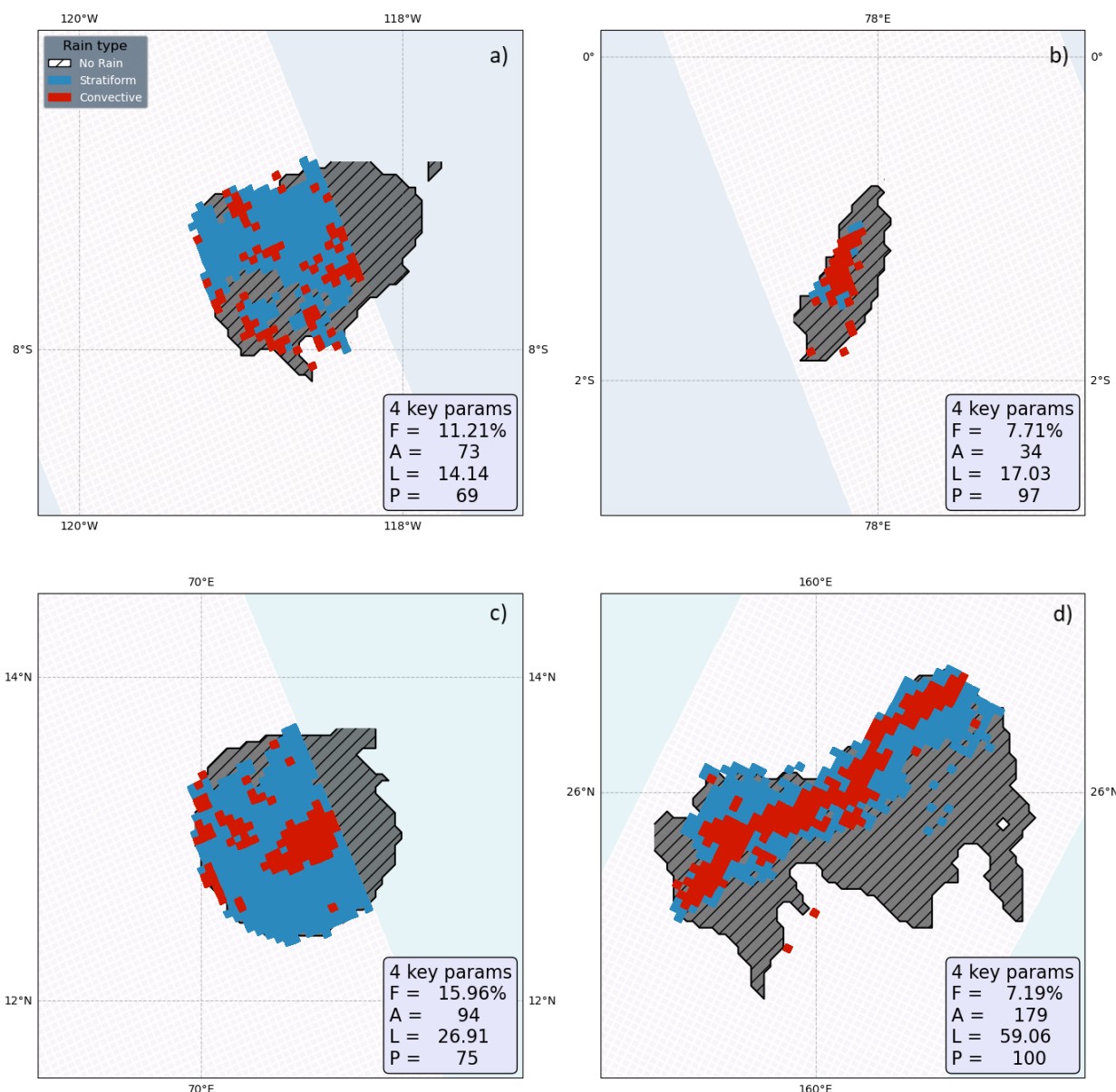


*Figure 14: Same as figure 1. Archetypal examples of a) likely random (LR), b) clustered small (CS), c)*
*clustered large (CL), d) very structured scenes (VS). The values of the four key parameters are indicated*
*for each example.*
The reprojection of the four classes onto the morphological aspects of the associated DCSs is illustrated
in Figure 15. As the level of organization increases from LR to VS, the distributions of these two
morphological parameters shift towards higher values, reaching a maximum for the VS, which are
predominant in the long tail (exceeding 20 hours and 80,000 km²). This finding indicates that the
organization, as measured by this approach, plays a substantial role in shaping the morphology of the
DCSs throughout their lifecycle.

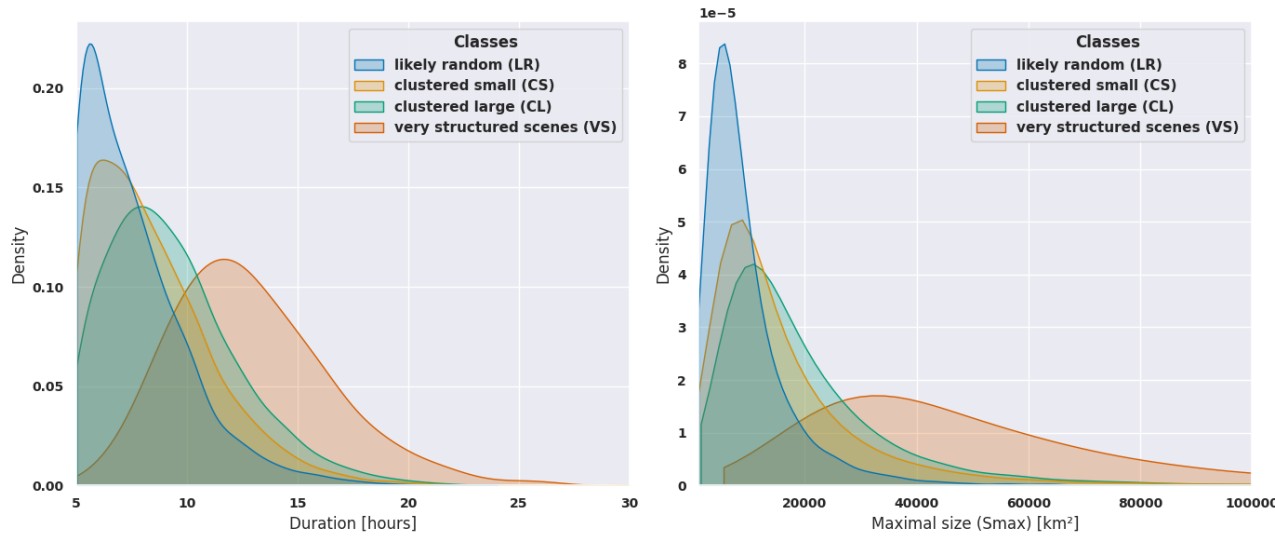

*Figure 15:* *Empirical probability density function (estimated via KDE) for each class from the TOOCAN-radar dataset. Left: Duration (lifetime in hours), Right: Maximum extension of the associated anvil cloud along the life cycle of the DCS.*

## 4 Discussion

The results above demonstrate that F and A alone, among the four parameters, are insufficient to effectively distinguish between organizational classes, as they only partially capture the spatial arrangement of convection. To comprehensively characterize the diversity and complexity of spatial patterns, additional diagnostic measures are necessary. We therefore introduced the more elaborated parameters L and P, which extend the analysis by incorporating both the spatial configuration of the elements and a estimation of the probability that the observed distribution deviates from randomness. These parameters offer a more nuanced and robust understanding of spatial organization beyond what F and A can provide.

As stated in the introduction, a number of metrics have been introduced to quantify the organization of convection (Biagioli and Tompkins 2023, Mandorli et al. 2024). Their ability to discriminate between the scenes has been already questioned and is further assessed using our two datasets as a benchmark. The likely random (LR) and very structured (VS) classes are clearly identified in both datasets and correspond to distinctly different arrangements of convection and are hence used to perform this assessment. Three well-known metrics are compared to our 2 most contrasted classes in Figure 16.

The first metric is the Convective Organization Potential (COP) index, introduced by White et al. (2018). COP assumes that larger and more closely spaced convective cores are more likely to interact, thereby contributing to organization. It yields a dimensionless value between 0 and 1, representing the degree of organization within a grid cell, by integrating the equivalent radii of cores and the pairwise distances between all cores. The second one, the Area-Based COP (ABCOP), is an adaptation of the original COP proposed by Jin et al. (2022). ABCOP enhances the COP framework by incorporating additional dependencies on the total area and the number of convective cores. The third diagnostic is the Radar Organization Metric (ROME), developed by Retsch et al. (2020). Unlike COP and ABCOP, ROME is expressed in surface units, with values ranging from one to two times the mean area of the convective cores. All three metrics share a common foundation in pairwise core analysis within a defined spatial domain.

Overall, these 3 metrics provide no clear separation between our two classes. The COP index exhibits an inverse trend, with highly structured scenes displaying lower COP values (Figure 16a, d). Among the three metrics, ROME provides the least effective class separation, particularly for the TOOCAN-radar dataset (Figure 16c, f). In contrast, the best separation is achieved using the ABCOP index (Figure 16b, e), especially in the TOOCAN-RCE dataset, although significant overlap remains in the central value range, particularly around 5 (unitless). ABCOP and ROME primarily reflect the total and mean convective object area, respectively (Mandorli et al. 2024), which corresponds to a combination of two of our four key parameters (F and A). This partly explains the effective separation of the two most contrasted classes by these indices. However, the fact that ABCOP distinguishes the classes more effectively than ROME suggests that organization is more strongly influenced by the total convective area (A) than by the mean convective object area. Nevertheless, the overlaps within ABCOP distributions indicate that the complexity of spatial arrangements cannot be fully captured by only A or even by its combination with F. This suggests that in order to achieve a more comprehensive distinction between different organizational patterns, spatial arrangement (L and P) has also to be considered.

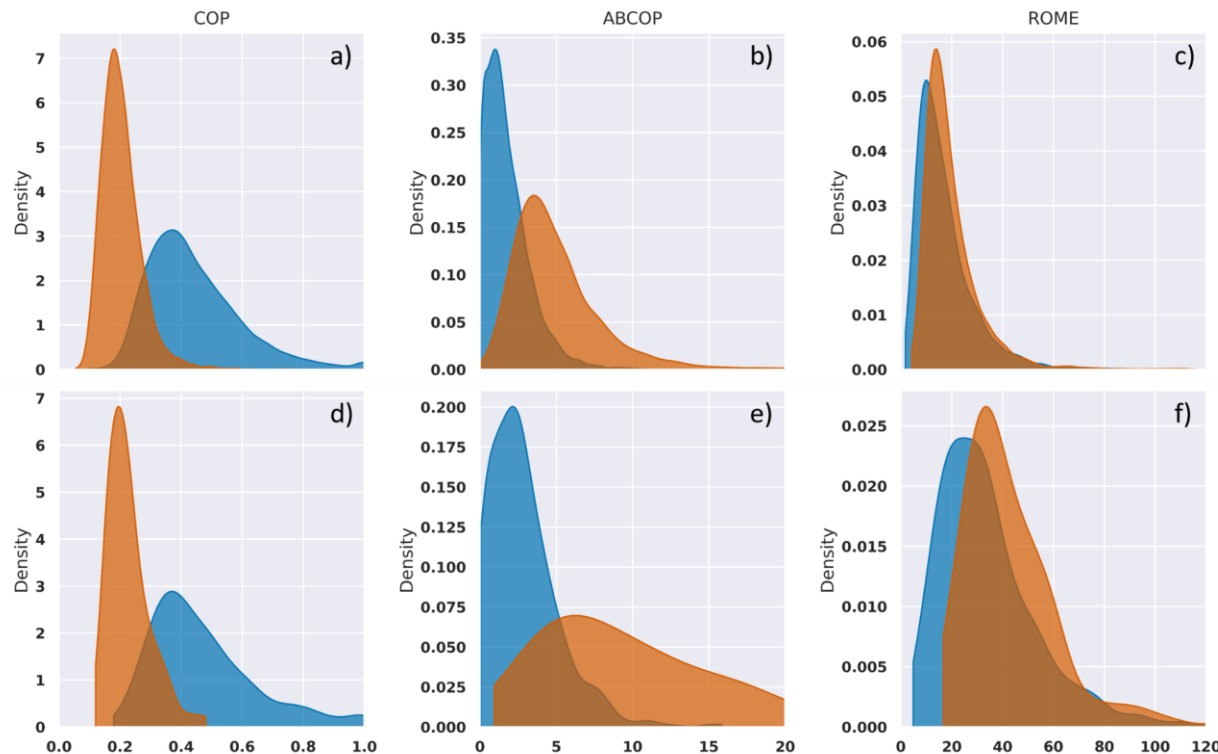

*Figure 16:* *Empirical probability density function (estimated via KDE) for the two extreme classes Top:*
*13443 scenes from TOOCAN-radar dataset, Bottom: 1055 scenes from TOOCAN-RCE dataset. a, d) COP*
*index, b, e) ABCOP index, c, f) ROME index. Likely random (LR) are represented in blue while very*
*structured (VS) scenes are colored in dark orange*

Thus, attempting to define a single index to quantify convective organization seems very difficult, as convective structures exhibit complex spatial arrangements that cannot be fully captured by a single metric. A low dimensional characterization (e.g., the 4 key parameters presented in this study) of convective organization is probably more appropriate as long as a comprehensive index that fully encapsulates this complexity has not yet been identified in the literature.

## 5 Conclusion

Organization of deep convection is crucial to better understand the role of deep convection in the climate system (Muller et al. 2022). In this study, a new method is introduced to quantify and characterize spatial arrangement of convective areas within a specific grid encompassing the instantaneous cloud shield of a deep convective system (DCS). This method extends previous works by i) considering a Lagrangian perspective, specifically by focusing on the spatial arrangement of deep convective cores within the cloud shield of the associated deep convective systems, ii) extracting a characteristic scale of convective core distribution based on the spectral power of the scene computed via 2D autocorrelation, and iii) computing a statistic ensemble of convective scenes based on elementary decomposition and stochastic random generation, establishing a probabilistic distance from a random spatial organization. This distance, expressed as a percentile (P), completes the characterization of each scene alongside its convection density (F), convective area (A), and characteristic length (L). This subset of 4 parameters describing deep convective organization aligns with the work of Janssens et al. 2021 suggesting that an effective low

dimensional characterization of organization can be achieved. In this work, the two more advanced
parameters L and P extend the analysis beyond F and A by quantifying the spatial configuration of the
cores and its deviation from randomness.
The robustness of the methodology is assessed by applying it to two independent datasets with distinct
convective core identification techniques, derived from satellite observations and kilometer-scale
numerical simulations. An unsupervised clustering approach is then employed to delineate four physically
sound organizational classes in both datasets. The resulting classification exhibits high consistency across
both datasets and quantifies the previously highlighted ambiguity found in existing organizational metrics
(Mandorli et al. 2024, Biagioli and Tompkins 2023).
The present approach has demonstrated good flexibility, effectively handling datasets with diverse scene
characteristics, including variations in size, convective fraction, and object count, while minimizing
arbitrary parameter choices and dataset dependencies, aligning with recent methodological
advancements (e.g., Koren et al., 2024). As convective organization spans a wide range of scales, from
isolated convective cores (≈5 km in diameter) to self-organized mesoscale convective systems with
diameters exceeding 100 km, exploring the scale independence and broader applicability of the present
method to such various fields or extending it to vertical velocities of updrafts in simulation or vertically
integrated precipitation from ground-based phased array radars would be one further promising
implication of this work. For instance, an adaptation to the spatial arrangement of DCS cloud shields within
a selected area, following Bony et al. (2020), could be considered using TOOCAN's segmentation as the
binary mask. However, this would require modifications, such as replacing (i) the elementary core
decomposition with convective system segmentation and (ii) the random generation process with
randomized positioning and orientation of them within the grid.
Examples of the potential applications of this method in weather and climate research include severe
weather and climate modelling. Establishing a statistical relationship between flooding (or strong wind
events) and the internal organization of convective systems would help nowcasting of these events by
completing the ingredients list (Doswell et al., 1996). Recent intercomparison exercises (Prein et al., 2024;
Feng et al., 2025) have revealed biases between the latest generation of km scale global models and the
satellite observed cloud shield morphology. The interpretation of these biases would benefit from
applying our method to these simulations to relate the limitations of the simulated morphology to that of
the convective organization as shown here.
One other venue for future work would be to extend the organization characterization with a time
dimension for datasets where this dimension is available such as ground-based radars and high-resolution
simulations. By incorporating time evolution of a selected field as a third dimension, the methodology
could assess spatiotemporal convective organization. Expanding the current 2D approach to a 3D
formulation will allow for a refined characterization of the temporality of convective organization,
reaching the individual dynamical convective cores life cycle scale in line with the work of Kim et al 2012
and Tseng et al. 2024)
In this study, specific methodological choices have been made regarding i) the elementary decomposition
into circular convective cores, ii) the stochastic generation model used to generate a distribution of
random scenes for quantifying deviations from randomness. One way to improve the first one would be
to work on the detection of the objects by implementing local maximum and water shading (or any other
region-growing techniques (Fiolleau et al. 2013) to a continuous field like precipitation for instance). The

second point could be enhanced by exploring other generating processes with different random probabilistic laws as the homogeneous Poisson point process (HPPP) (e.g., Savre 2024). In particular, these refinements will help understand the underlying stochastic model governing organized convection and will require dedicated work with a more systematic framework such as the one of Biagioli (2023).

Additionally, based on the TOOCAN-RCE dataset, or any dataset providing continuous fields across all time steps, it becomes feasible to infer the temporal evolution of the four organizational parameters: F(t), S(t), L(t), and P(t). By tracking these metrics as functions of time, one can characterize the dynamical evolution of the convective system's organization within the four-dimensional space. This time-resolved approach enables a detailed assessment of how the degree and nature of organization evolve throughout the lifecycle of the DCS, offering new insights into its developmental phases and potential transitions between different organizational states.

Finally, the observational classified dataset is now being used to explore the relationship between different organization patterns and the morphology of the associated convective systems. On-going physically driven analysis shows promising results whereby the "organized" systems exhibit a scale dependence sensitive to the depth of convection and will be further explored in upcoming studies.

**Acknowledgments**

We thank S. Cloché for her support with the handling of these various datasets. This study benefited from the IPSL mesocenter ESPRI facility which is supported by CNRS, UPMC, Labex L-IPSL, CNES and Ecole Polytechnique. The authors acknowledge the CNES and CNRS support under the Megha-Tropiques program.

**Open Research**

The TOOCAN data are available from **http://toocan.ipsl.fr** with the with DOI https://doi.org/10.14768/1be7fd53-8b81-416e-90d5-002b36b30cf8

The 2012-2020 homogenized infrared geostationary level-1C dataset described in this paper can be accessed via the repository under the following data DOI: https://doi.org/10.14768/93f138f5-a553-4691-96ed-952fd32d2fc3 (Fiolleau and Roca, 2023).

Toshio Iguchi, Robert Meneghini (2021), GPM DPR Precipitation Profile L2A 1.5 hours 5 km V07, Greenbelt, MD, Goddard Earth Sciences Data and Information Services Center (GES DISC), Accessed: 10.5067/GPM/DPR/GPM/2A/07

The simulation data, along with the associated analyses, are currently in the process of being registered and will be identified with a DOI to ensure persistent accessibility and citation.

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

## Annex A: Decomposition into elementary cores:

**Decomposition into elementary cores: methodology description**

The decomposition process involves two main steps. The first is to find the maximum size of a square cluster that can fit within the mask: MaxSquareSize (Figure A1, middle). The algorithm scans the grid from top to bottom and left to right, assigning to each pixel the sum of its own value and the minimum value among the three neighboring pixels: the one above, the one to the left, and the one above-left. This step is illustrated in Figure A1, middle. The second step labels the structures based on the sizes identified in the first step, filling them from the largest clusters down to the 1x1 ones (Figure A1, right). As stated in the main text (see section 2.2), this partitioning algorithm could be non-unique, occurring in approximately 10% (TOOCAN-radar dataset) and 25% (TOOCAN-RCE dataset) of all scenes. This arises from the direction in which the partitioning algorithm scans the image pixels. However, a sensitivity analysis (not shown) revealed that this non-uniqueness has no significant effect on the subsequent methodology. There is no significant modification of the generated L distribution and therefore in the computation of the P value. Besides, the classification results remain consistent for over 99.6% of affected scenes in the TOOCAN-radar dataset and 95.6% in the TOOCAN-RCE dataset.

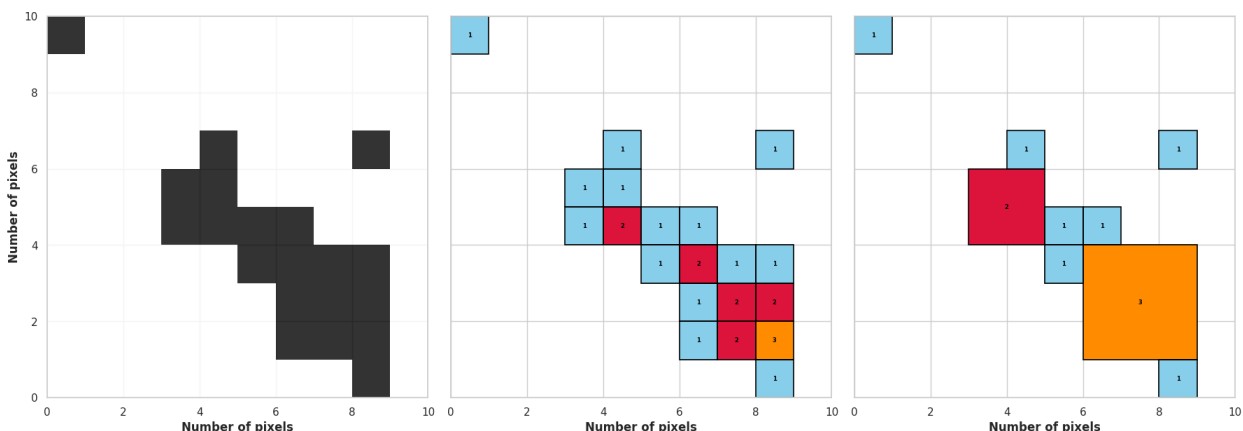

*Figure A1:* Left: 2D binary scene (convective mask). Middle: Identification of the maximum size (MaxSquareSize) within the grid before decomposition. Right: Elementary decomposition into square clusters.

## Annex B: Sensitivity

**Sensitivity to selected parameters**

### Threshold for computation of the autocorrelation characteristic scale

The 10% threshold T is modified to account for the sensitivity to this parameter. With T=5% and T=15%, the results for the L values differ slightly as the scale of the contour is directly impacted by this value. Figure B1 shows the realizations of 500 generated L for various T thresholds. As T increases, the L values tend to decrease generally, but the P values do not vary much once compared to the generated L values obtained with the same T value. The major impact of T (and so its value definition) is on the L range but do not have major consequences in the classification process (e.g., Kmean used with normalized data)

For our usage in both convective areas from spaceborne radar and convective regions in simulations, the sensitivity study indicates that the 10% threshold remains the most suitable for encompassing a large diversity of scenes with a range of convective fractions between 0.5 to 25% without producing reasonable doubts in the classification process. It is noteworthy to be cautious for usage with convective fields that often count for more than 25% with large (up to 5 and more in size length) square elementary cores. We suggest that it could be interesting in such cases to increase the contour threshold up to 15% to better capture a proper range for L values.

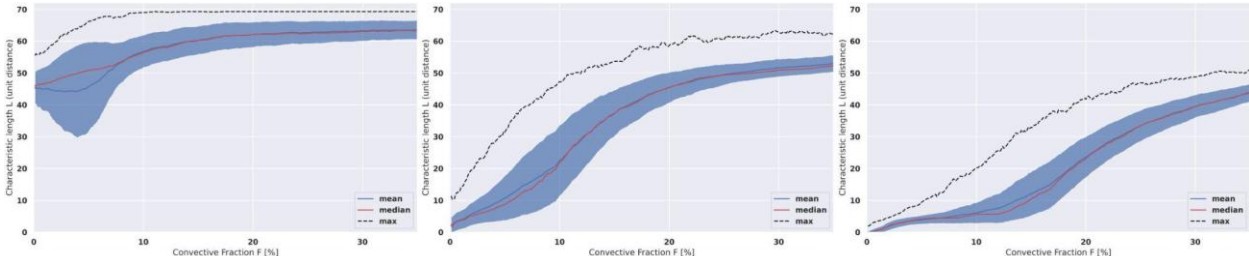

*Figure B1:* Same as figure 10 with Left: T=5% threshold, Middle: T=10% threshold, Right: T=15% threshold for square clusters partition with MaxSquareSize <= 3.

### Monte-Carlo approach for generating a random reference distribution

The same sensitivity study was conducted for the parameter regarding the amount of each Monte-Carlo generation of grids, with the number of generations being large enough to make a robust distribution. We came to the number of 500 as a perfectly suitable number of generated scenes for each real one, as a good compromise between computation time and statistical liability. With 500 iterations, the generated distributions of characteristic lengths exhibit a wide range of values with relatively smooth distributions, often exponential-like or with no common shape (Figure 9). The computation of P is always feasible, independently of the distribution shape.

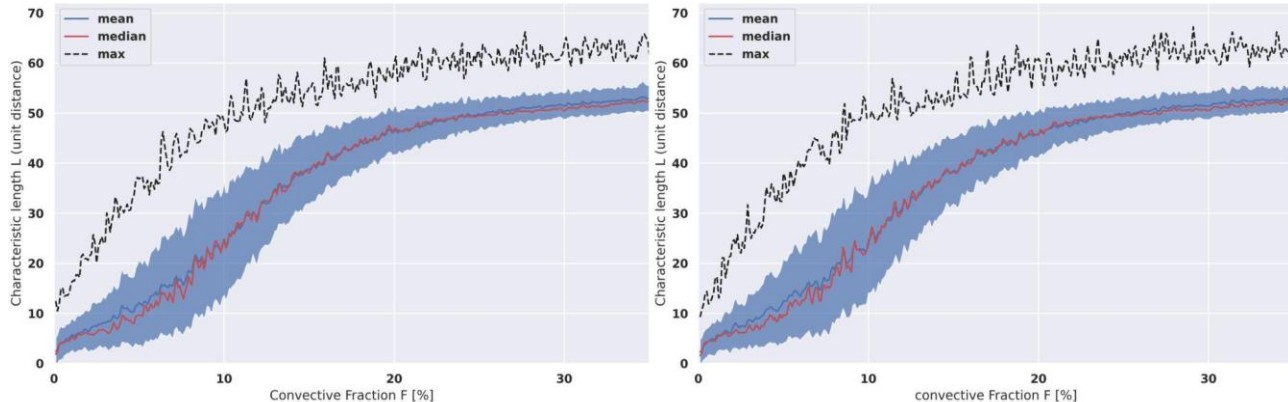


*Figure B2: Same as figure 10 with Left: 100 generations, Right: 1000 generations, for square clusters*
*partition with MaxSquareSize <= 3.  Here there is a 0.5% rolling mean applied for readability and to allow*
*comparison.*
## Monte-Carlo-style error estimation within the clustering algorithm
A Monte-Carlo-style study was conducted on K-means clustering, adding noise to each parameter in the
radar dataset (~60,000 points). For each parameter, we added uniform noise with amplitude [-
alpha/100*(Perc_90-Perc_10), +alpha/100*(Perc_90-Perc_10)], with alpha in % for values [0.5, 1, 2, 5, 10,
15, 20, 30]. Then we reclassify using the same Kmean algorithm (under the same conditions), and compare
the class labels, calculating the number of points that change classes (or their proportion), the change in
the silhouette score, and the Adjusted Rand Index (ARI), which quantifies the agreement between two
partitions, correcting for random fluctuation. An ARI close to 1 means a very high correspondence
between the two partitions, and when it approaches 0, we are close to an agreement corresponding to
random draws. We perform this process 10 times for each value of alpha to estimate uncertainty on the
mean values of each of the metrics, for each of the parameters. The results are presented in the figure
below (that corresponds to the added Figure B3).

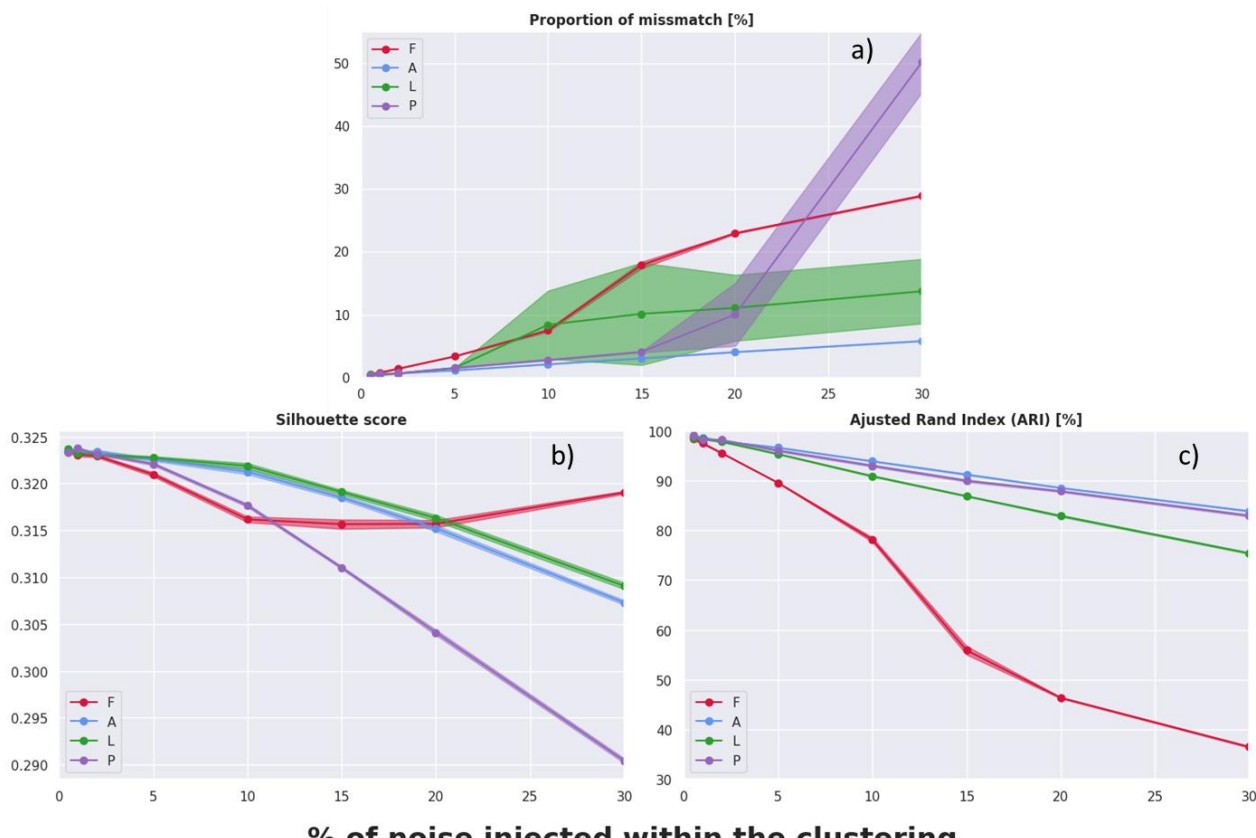


 *Figure B3: Variation in a) proportion of mismatch, b) Silhouette score, c) Adjusted Rand Index (ARI),*
*comparing classification performance with perturbation by adding noise to the distributions of the four*
*parameters (convective fraction (F), convective area (A), characteristic length (L) and percentile (P)*
*shown in color), as a function of the percentage of noise added. The shaded areas correspond to the*
*uncertainties for each measure (dots) computed as +/- 1 σ/√10 (10 being the number of carried*
*experiments for each of the values).*

This experiment allows to quantify systematic disturbance/error on each parameter and then define an order of importance regarding parameter sensitivity in the clustering process to perform classification. Based on the 3 metrics in Figure B3, A is the less impactful parameter when disturbed with noise, followed by P and L (depending on the index) when the percentage of noise is less than 15% which is already a significant change. Then F appears to be the dominant parameter, showing globally the highest (respectively lowest) scores in Figure B3a (resp. Figure B3b, c). Regarding the silhouette score, P becomes the dominating factor for alpha values higher than 15%, meaning that over this threshold for P, the centroids' position of the 4 classes become unstable. Overall, this demonstrates that up to +/- 5% noise on all parameters, the classification remains stable (with ARI>0.9, and even up to >10% for S, P, and L. Only F shows greater sensitivity above 10%. Moreover, the uncertainty estimated with 10 experiments for each of the selected set of parameters/alpha combination also shows that the method is statistically robust (shaded areas on Figure B3).