# Peer review of "A method for characterizing the spatial organization of deep convective cores"

_EGUsphere, 2025_

## Author Comment (AC1)

We thank the referees for their thoughtful and insightful comments which have helped us improve this manuscript.

In light of several converging feedbacks, we propose changing the title as follows: "A method for characterizing the spatial organization of deep convective cores in deep convective systems' cloud shield using idealized elementary convective structure decomposition", putting emphasis on the assumption of elementary convective segmentation with circular structures of varying radii.

Another major change is that the convective area S has been replaced by the letter A to align more closely with the term 'area' rather than 'surface'. In our response, we have kept S consistent with the comments made, but all occurrences have been modified within the manuscript.

We have also updated some of the figures as requested, along with adjustments in the text. Two additional figures have been added, the first one in the "Results" section (new Figure 15), and the last one in the Annex B section (Figure B3). In addition, we have reinforced the sensitivity study in Annex B by conducting a Monte Carlo-style experiment where we propagate error within the clustering process, in order to evaluate the stability of the method with respect to each of the four key parameters. Text has been amended in tracked changes.

Below we provide point-by-point responses to each comment.

**RC1 :**

This paper introduces a novel method to classify and study the spatial organization and geometry of convective systems as well as their distribution of convective cores. Four key metrics are introduced based on infrared brightness temperatures in geostationary satellite observations and outgoing longwave radiation in km-scale numerical simulations (with a delta x of 3km) of idealized cloud scenes. The metrics give a more holistic description of convective organization, including the size and relative fraction of convective cores as well as their spatial arrangement and randomness. The metrics are discussed well in the context of already existing metrics for convective organization and the authors make an argument that convective organization cannot be well described when relying on one single parameter instead of combining different aspects of the spatial structure. Overall, the paper is well-structured, well-written and is a valuable contribution to allow for a more sophisticated analysis of convective processes. While I do not think that any additional analyses are needed, I think there are still a few parts in the paper that need clarifications, in particular because the main point of the paper is to introduce a new technique that should be straightforward to reproduce. I recommend the paper for minor revisions.

We would like to thank the referee for the positive and constructive feedback. We will address all the points in the following. Please see below our responses to the individual comments, with the original comments included in black and answers in blue.

General comments:

Physical processes

Acknowledging that this paper focuses on the introduction and development of a new method, it is not expected that the conclusions center around new findings about convective organization. However, I think the paper would benefit from explaining how the introduced metrics and the classes that are based upon these four metrics, could be used to identify certain processes. Right now, the differences in the distributions (e.g. in Fig. 13-14) are not really discussed in the context of which underlying weather systems and processes would produce the different signatures in convective organization.

We would like to point out that the main purpose of this paper is not to explore internal processes, but rather to constrain the impacts of the organization of convection in relation to the morphological properties of thunderstorms.

We have added a figure in the results section showing for each of the classes the distributions of two key morphological parameters of the DCS life cycle: Duration (lifetime) and maximum extension of the associated anvil cloud (Smax). The results are discussed in a paragraph added at the end of the section.

[Figure]

*Figure 15: Empirical probability density function (estimated via KDE) for each class from the TOOCAN-radar dataset. Left: Duration (lifetime in hours), Right: Maximum extension of the associated anvil cloud along the life cycle of the DCS.*

This said a direct way to explore the different processes that would produce different organization of convection would be to extend a recent study we published (Roca et al., 2025) where the dynamical and thermodynamical environment in which the system morphology evolves has been analyzed. For instance, it would be interesting to link the F and C parameters to CAPE and precipitable water while the L and P might be more related to the vertical shear of the horizontal wind. Such efforts are underway and will be part of another study.

In addition, it would be useful to name a few examples of how this method can be applied in weather and climate research.

We have added the following paragraph to the conclusion section:

"Examples of the potential applications of this method in the realm of weather and climate research include severe weather and climate modelling. Establishing a statistical relationship between flooding (or strong winds events) and the internal organization of the system would help nowcasting of these events by completing the ingredients list (Doswell et al., 1996). Recent intercomparison exercises (Prein et al., 2024; Feng et al., 2025) have revealed biases between the latest generation of km scale global models and the satellite observed cloud shield morphology. The interpretation of these biases would benefit from applying our method to these simulations to relate the limitations of the simulated morphology to that of the convective organization as shown here."

Decomposition into elementary structure

It appears to me that the decomposition into the elementary structures that is based on the square-like nature of the convective core has implications for the characteristic length L. Can you explain in more detail if this assumption is a limitation for the range of L values that you get?

This point was also raised by the second reviewer. First, we have changed the title to highlight the use of this strong hypothesis as stated before by adding "using idealized elementary convective structure decomposition". Next, we assume that the referee requests details on the influence of the elementary decomposition on the P parameter rather than the L parameter, since the L parameter is independent of it. Only the L distribution of the generated scenes may be influenced by the assumptions of the decomposition.

Firstly, autocorrelation is sensitive to dominant structures. Using structures composed of only one pixel is often not discriminating for low fractions (F). Figure 10 (left) shows the range of variation of L for a given F in a 50x50 grid with randomly positioned 1x1 structures. For low F values, the characteristic length values are all confined until F exceeds a certain threshold. The same is true when 2x2 structures are also permitted, with the range of F expanding more rapidly. The same pattern continues up to 5x5 structures (Figure 10, right). See Figure below that complements the Figure 10.

[Figure]

*Same as figure 10 with Left: only square clusters of size 1x1 and 2x2 (MaxSquareSize = 2), Left: only square clusters of size 1x1 and 2x2 and 3x3 (MaxSquareSize = 3).*

Besides, it is impossible to break down all scenes using only structures that are 2x2 or larger. Almost always, structures that are 1x1 must be used in addition.

The elementary decomposition impacts the calculation of L distribution for generated scenes, with the majority of the influence coming from the segmentation of dominant structures (larger squares or larger objects if a different decomposition is used). In fact, the more roughly the structures are cut up, the more sensitive L is to the main structures and their respective arrangements. Conversely, by cutting everything into finer elements, for example into 1x1 and/or 2x2 objects, which is also always possible, L will have a more confined range of values and will therefore struggle to express itself in a sufficiently broad distribution as a key parameter for the rest of the characterization process. We therefore have a much greater impact on P than on L itself here, which we have quantified in the example using the figure 3:

[Figure]

*Same as figure 3 (left and right only) with decomposition only with b) square clusters of size 1x1, c) size 1x1 and 2x2 (MaxSquareSize = 2), d) size 1x1, 2x2 and 3x3 (MaxSquareSize = 3)*

[Figure]

*Same as figure 9 (left only) with decomposition only with a) square clusters of size 1x1, b) size 1x1 and 2x2 (MaxSquareSize = 2), c) size 1x1, 2x2 and 3x3 (MaxSquareSize = 3).*

When decomposing with either 1x1 or 1x1 and 2x2 structures, P takes on higher values, suggesting that the organisation is less random. The L distribution shifts to higher values when using larger structures. In other words, the inclusion of large elements augments the probability for the final structure to occur from pure randomness compared to not including larger structures.

Differences between observations and model simulations

In the very beginning of the paper, the definition of "convective" vs. "stratiform" is introduced for models and observations. I am wondering how much of the differences that you find between the datasets actually go back to how you define "convective" in the model dataset. It makes sense to base this definition on physical processes such as updrafts, but I am afraid that, as you mention yourself, this would lead to significantly different regions than the purely radar-based signature in the satellite observations.

There is some ambiguity as to the definition of a convective column. In order to assess the sensitivity of our method to this ambiguity we have used two datasets each with a different definition. But we are not elaborating on the differences between the model and the observation or between the definitions.

In addition to that, I think that the paper would benefit from a more thorough discussion of differences between models and observations, as an example for how to apply the introduced method (i.e. that it can help to validate models that partly resolve convective processes).

While we are definitely interested in the suggestion, the focus of this paper is primarily on the impact of convective organisation on the cloud shield of DCSs. Therefore, the aim of this study is not to validate or compare models with satellite observations as such a comparison is not straightforward and would require a dedicated effort beyond the scope of the present paper.

Figure labels

The font size in all figures need to be increased. In some figures, such as, Fig. 3, the labels are barely readable.

Done, most of the figures/labels have been enlarged

Detailed comments

21: Variables -> metrics ? (variables sounds more like specific atmospheric fields whereas metrics specifies that you introduce and suggest a measure)

Yes, we decided to modify all the occurrences of "variables" by "parameter".

27: *Idealized* kilo-meterscale numerical simulation

Done.

54 - 67: When reviewing the existing literature on convective organization indices, could you go a bit more into detail on what variables these indices are usually based on?

Yes, we added this sentence to give more specific context:

⇒ "Other existing metrics that are more object oriented such as COP, ABCOP and ROME (White et al. (2018), Jin et al. (2022), Retsch et al. (2020)) consider different convection attributes, like the object's area, mean area of objects, relative or geometric distance between objects, in addition to their center within a specific grid."

   1. 104: remove "…"

Done.

L,. 134 ff: When introducing the radar collocations, could you clarify which variables you are working with - is it only the retrieved rain rates or other retrieved quantities? Do you leverage the radar reflectivity values at all?

Here it is only the precipitation type classification that is at stake. Other variables available will be used in future works. All the details can be found in the following references (and the ones that were already cited):

*Le, M. and V. Chandrasekar, 2013a: Precipitation Type Classification Method for DualFrequency Precipitation Radar (DPR) Onboard the GPM, IEEE Trans. Geosci. Remote Sens., 51(3), 1784–1790.*

*Awaka, J., T. Iguchi, and K. Okamoto, 2009: TRMM PR standard algorithm 2A23 and its performance on bright band detection, J. Meteor. Soc. Japan, 87A, 31–52.*

We precise a bit as follow, adding the two references cited above:

⇒ "For our analysis, we focus on the classification of precipitation type into stratiform, convective and other rain type, retrieved from the 2A23 algorithm (Awaka et al. 2009, Le and Chandrasekar 2013, Awaka et al., 2016, 2021; Chen et al., 2025)."

144, 146: colocation -> collocation

Done

179: Is this classification based on radar reflectivity thresholds?

The classification process for 2A23 is very complex, as described in details in its ATBD: https://gpm.nasa.gov/sites/default/files/document_files/ATBD_DPR_201811_with_Appendix 3b_0.pdf

We modified the incriminated part as follows: ⇒ "Within the radar swath, precipitation is classified as stratiform, convective, or non-precipitating, following the precipitation type from the 2A23 algorithm introduced before, and excluding the "other" rain type. Convective echoes are highlighted in red in Fig. 1."

1. 236: It makes sense that a grid cell can be classified as convective either when it contains heavy convective rainfall or when there are updrafts that indicate strong vertical motion (which may happen prior to and in a different grid cell than the heavy precipitation). However, it is not clear to me why a grid cell below the downdraft threshold would also be classified as convective. While strong downdrafts are part of the convective system as a whole, the processes are quite different than in updraft regions, so I would like to better understand what we gain from having strong updrafts and strong downdrafts in the same category. Would it make sense to leverage the information of hydrometeors in the model?

We thank the referee for pointing these out. We have not properly introduced the method of Marinescu and we shortcut way too much the methodology in the original manuscript.

Indeed the two thresholds correspond to two regions of the column and consider the absolute value of the vertical velocity (Marinescu et al., 2016). So it either can be updrafts or downdrafts. The fact that both can be classified as "convective" can be understood as being the two sides of a convective cell with a given life cycle as shown on the following figure of a mature convective cell (Markowski and Richardson, 2010). See also the marinescu et al paper and references therein.

[Figure]

*A mature convective cell. from Markowski, P., and Y. Richardson, 2010: Mesoscale Meteorology in Midlatitudes. Wiley, 1–407 pp.*

The original sentence "be also classified as convective if the vertical velocity above the 0°C isotherm exceeds the updraft threshold or falls below the downdraft threshold." as been modified in to: "be also classified as convective if the absolute value of vertical velocity updraft (or downdraft) above (below) the 0°C isotherm exceed 5 m/s (3m/s)"

Table 1 also is modified and now reads:

| Convective Precipitation threshold (mm/h) | 0°C isotherm height (m) | Cloud top height (m) | stratiform Precipitation threshold (mm/h) | Cirriform OLR threshold (Wm-2) | Updraft threshold above the 0°C isotherm (m/s) | Downdrafts threshold below the 0°C isotherm(m/s) |
|---|---|---|---|---|---|---|
| 10 | 3000 | 6000 | 0.04 | 172 | 5 | -3 |

Note that the algorithm was originally correctly implemented though.

1. 286: Can you explain the "high-resolution precipitation ground based core and updrafts/downdrafts joint occurrence analysis"?

The two mentioned papers (Moroda et al.; Lamer et al., 2023) report case study analyses of high resolution vertically resolved simultaneous measurements of both the dynamics and the precipitation within storms using Phased Array Weather Radars. These analyses reveal the interplay between the hydrometeor distribution and the vertical velocity, and emphasize the strong but complex relationship that takes place at small space and time scales. Moroda et al. build a conceptual model of the lifecycle of the precipitation cores and the updraft trajectory in the precipitation cell. Their analysis is based on a joint occurrence analysis of precipitation and updrafts/downdrafts.

To clarify our purpose the incriminated sentence is modified and now reads "precipitation core and updrafts/downdrafts joint occurrence analysis from high-resolution ground based observations."

1. 282- 284: I understand the assumption that the hydrometeor distribution is the result of small-scale convective dynamics and cloud microphysics, and that, in other words, we can see convective cores as aggregates of these small-scale processes.

   But where does the assumption of a square-like geometry come from? Could it not be that these small-scale aggregates are, for instance, elongated like in squall lines, or is there evidence that convective cores have a circle- or square-like geometry with the same distance to all its surroundings?

The circular assumption relies on a vertically erected plume model which is a useful idealization of deep convection dynamics. In this simple case, the shape of the cross section of the plume is indeed circular. Yet if the plume is slanted (due to wind shear), the cross section of the plume would look like an ellipse. The circular assumption (which translates into square pixels) should be understood as an idealized, first order approximation. To better convey this dimension of our work the title of the paper has been modified.

Fig. 4: Is the core size given in the number of pixels?

⇒ Figure 4 caption was modified : "Size distribution of the convective cores Left: TOOCAN-radar dataset. Right: TOOCAN-RCE dataset. The Houze-like detection is shown in blue and corresponds to the rounded integer value of the square root of the cores' area, while our elementary decomposition is shown in red and corresponds to the size of the square cores."

Fig. 5: Add in the figure caption that this is for the satellite data. Are those the composites over all tracked DCSs?

We modified it as follows in the text ⇒ "Figure 5 shows a comparison of the mean vertical profile of reflectivity for the present decomposition and that of Houze-like cores, using 8 years of collocated radar measurements to allow for robust statistics."

And in the figure caption: "Figure 5: Reflectivity profiles (composites using 8 years of collocated radar measurements) of convective cores for Left: our elementary decomposition, Right: Houze-like shared edges decomposition, with different core sizes shown in color".

1. 657: A new method for what?

The incriminated sentence was reformulated: "In this study, a new method is introduced to quantify and characterize spatial arrangement of convective areas within a specific grid encompassing the instantaneous cloudshield of a DCS."

1. 727: Please check the DOI of your dataset. I could not access it.

We checked the DOI of our datasets and it seems to work from our side:

TOOCAN dataset: https://doi.org/10.14768/1be7fd53-8b81-416e-90d5-002b36b30cf8

IR GEO-ring dataset: https://doi.org/10.14768/93f138f5-a553-4691-96ed-952fd32d2fc3

1. 344: What is meant by this sentence: "As a consequence, unlike regular gridded data, the method is required to be shield-specific." I am confused by the statement that the method cannot work for regular gridded data.

Thank you. We have modified the incriminated sentence into : "As a consequence, unlike regular gridded data of fixed size, the method requires defining a selected area that corresponds to the cloudshield dimensions".

1. 371: It is not clear what the difference is between the "distribution of convection across the scene" and its "spatial arrangement". As I understand it the first two metrics describe the area and amount of convection ("how much?") and the last two methods focus more on the geometry and where within the cloud shield convection is present?

We modified it as follows: "The first two parameters describe the amount of convection within the scene, using both absolute and relative metrics."

Fig. 11: The difference in the maximal area between the satellite data and the model simulation appears to be quite significant. Is this a consequence of the effective resolution of the datasets, and does this indicate that the model physics cannot reproduce the large cloud shields that we can observe? In addition, the duration of the DCs seems similar, which is interesting because the spatial and temporal scales should also be linked or not?  I think it would be useful to discuss this point in more detail.

It is not possible to make a one-to-one comparison between the real world and idealized long-channel RCE (no Coriolis) simulations, for which the scales are not directly comparable with reality. So, it is not a problem of the model physics, but a problem of the numerical experiment set-up.

1. 544: I think it would be helpful to remind the reader what the four key variables are in the figure caption?

Done in figure caption.

L. 546: If I understood it correctly, K-means clustering is used to produce the four classes in Fig. 13 and these classes are based on the multivariate coherence of the four metrics.

Yes it's totally correct.

It is, however, not explained in detail how the PDFs of each metric relate to the respective class (from random to organized). For instance, Fig. 13 a) shows quite distinct distributions for F between class 0 and 1 although these are the classes closer related to each other. I do not expect to go into the details of all possible combinations of the four metrics, but it would be decent to describe a little bit more how convective systems with substantially different distributions for F and P can still be more alike each other when they are similar in terms of L and S.

Yes indeed. This fact emerges from the inherent differences between class 0 and the rest, as class 0 corresponds to a more likely random distribution of the cores, while the three others have higher values in the P distribution. The classification from random to organized is based on the P distribution (and secondary on the A and L variables for which the order is the same in terms of means and modes of distribution), that is more and more close to 1 as the class number increases. Class 0 and Class 1 are not much closer to each other than any other Class, it is simply a distance between points in a 4-dimensional space that we are trying to interpret more physically.

⇒ We included a modification in the dedicated paragraph to express this point: "As P increases (and secondary L and A too), so does the level of organization."

---

## Author Comment (AC2)

The manuscript entitled "A method for characterizing the spatial organization of deep convective cores in deep convective systems' cloud shield" presents a comprehensive methodology for analyzing the spatial organization of deep convective cores (DCCs) within Mesoscale Convective Systems (MCSs). Recognizing the limitations of existing organization indices when used in isolation, the authors propose a multidimensional approach based on four variables: (1) the convective fraction, (2) the total area of deep convective cores, (3) a characteristic length scale of aggregation among the cores, and (4) a metric quantifying deviation from a uniform spatial distribution.

A notable innovation of the study lies in the representation of DCCs as filled squares within the deep convective region, which enables the characterization of the spatial organization of the most elemental deep convective structures, although this identification relies on strong assumptions. Applying this framework to both observational and model datasets, the authors identify four distinct modes of DCC organization. They conclude by emphasizing the method's effectiveness, robustness, and adaptability, and suggest several promising avenues for future application.

Overall, this manuscript is of good scientific quality and aligns well with the scope of Atmospheric Measurement Techniques (AMT). The scientific content is clearly presented and well written. However, I have a few minor comments that should be addressed prior to publication. My main concern pertains to the robustness of the proposed method, as the sensitivity analyses provided are, in my view, somewhat insufficient (see detailed comments below).

We thank the referees for their thoughtful and insightful comments which have helped us improve this manuscript. In light of several converging feedbacks on the importance of the decomposition process, we propose to change the title as follows: "A method for characterizing the spatial organization of deep convective cores in deep convective systems' cloud shield using idealized elementary convective structure decomposition", putting emphasis on the assumption of elementary convective segmentation with circular structures of varying radii.

Another major change is that the convective area S has been replaced by the letter A to align more closely with the term 'area' rather than 'surface'. In our response, we have kept S consistent with the comments made, but all occurrences have been modified within the manuscript.

We have also updated some of the figures as requested, along with adjustments in the text. Two additional figures have been added, the first one in the end of the Results section as the new Figure 15, and the last one in the Annex B section as Figure B3. In addition, we reinforce the sensitivity study in Annex B by conducting a Monte Carlo-style experiment on propagated error within the clustering process, in order to evaluate the stability of the method with respect to each of the four key parameters. Text has been amended in tracked changes.

And in particular we have emphasized the analysis of the robustness of the method. In addition, we reinforce the sensitivity study in Annex B by conducting a Monte Carlo-style experiment on propagated error within the clustering process, in order to evaluate the stability of the method with respect to each of the four key parameters. Our point-by-point responses are detailed below, with the original comments included in black and answers in blue.

Detailed comments

Figures:

The labels are sometimes too small. Besides, adding letters to identify each panel could help.

Done, most of the figures/labels have been enlarged

L. 138-140:

There are some words missing in this sentence.

The incriminated sentence has been modified as follows: "To ensure continuity, harmonization of TRMM-PR and GPM Ku-band (13.6 GHz) calibration, reducing systematic differences and improving long-term precipitation estimates." ⇒ "In order to ensure continuity, harmonization of TRMM-PR and GPM Ku-band (13.6 GHz) calibration has been carried out, with the objective of reducing systematic differences and improving long-term precipitation estimates."

L. 279-284: elementary convective structures

Could the authors clarify whether there is a physical justification for defining, for example, a 4×4 square as a single elementary convective structure rather than interpreting it as four 2×2 structures grouped together?

The continuity of the horizontal convective structure is inherited from the concept of ensembles of updrafts that are often observed aggregated to one another in radar observations, especially with relatively low spatial resolution (see early work by Houze and for instance Houze, 1997). This is also the case in airborne identification of updrafts over tropical oceans (see Zipser et al., 1994; Lemone et al., 1998). From this reflectivity (hydrometeors) or dynamical based definitions, a spectrum of continuous space scales has been observed and related to a dynamical core or a bulk plume. The homogeneous spatial distribution of the parameter is hence used to define the event. Details vary on how to define continuity (4 or 8 connectivity), on the threshold selection etc., but all techniques concur to identify a spectrum of horizontally spread objects with various sizes.

In this view, a continuous object would be a given convective entity. This is what we refer to as the Houze method in the manuscript. This classical approach does not imply any preferred morphology for the object. The present decomposition goes one step further by imposing a morphological constraint on the object delineation. A circular (square) shape is hence assumed and the most direct implementation of it is the method we have chosen: any continuous square object is associated with an individual core. That's why the method identifies a single 4x4 pixel core instead of 4 2x2 pixel cores.

The last sentences of the paragraph is modified as follows: "This approach can be viewed as an upscaled, space-borne version of the analysis of precipitation core and updrafts/downdrafts joint occurrence analysis from high-resolution ground-based observations (e.g., Moroda et al., 2021; Lamer et al., 2023). In this perspective, we assume that the large cluster of continuous echoes (colored in green in Figure 3, middle) is composed of smaller coherent and compact convective features akin to a circular bulk updraft of varying diameter that we approximate

using size varying squares. We hence add a morphological constraint to the classical core segmentation."

Consequences of this assumption are further detailed below.

In particular, how does this assumption impact the calculation of the spatial organization metric (variable P)? It would be helpful to discuss whether this structural definition influences the interpretability of P.

This point was also raised by the first reviewer, and we will structure our response around the shared concerns:

Firstly, autocorrelation is sensitive to dominant structures. Using structures composed of only one pixel is often not discriminating for low fractions (F). Figure 10 (left) shows the range of variation of L for a given F in a 50x50 grid with randomly positioned 1x1 structures. For low F values, the values are all confined until F exceeds a certain threshold. The same is true when 2x2 structures are also permitted, with the range of L expanding more rapidly with increasing F values. The same pattern continues up to 5x5 structures (Figure 10, right). See the figure below that complements the Figure 10, with generations with 1x1 and 2x2 permitted etc.

[Figure]

*Same as figure 10 with Left: only square clusters of size 1x1 and 2x2 (MaxSquareSize = 2), Left: only square clusters of size 1x1 and 2x2 and 3x3 (MaxSquareSize = 3).*

Moreover, it is impossible to break down all scenes using only structures that are 2x2 or larger. Almost always, structures that are 1x1 must be used in addition. Thus, the elementary decomposition impacts the calculation of L distribution for generated scenes, with the majority of the influence coming from the segmentation of dominant structures (larger squares or larger objects if a different decomposition is used). In fact, the more roughly the structures are cut up, the more sensitive L is to the main structures and their respective arrangements. Conversely, by cutting everything into finer elements, for example into 1x1 and/or 2x2 objects, which is also always possible, L will have a more confined range of values and will therefore struggle to express itself in a sufficiently broad distribution as a key variable for the rest of the characterization process. We therefore have a much greater impact on P than on L itself here, which we have quantified in the example using the figure 3:

[Figure]

*Same as figure 3 (left and right only) with decomposition only with b) square clusters of size 1x1, c) size 1x1 and 2x2 (MaxSquareSize = 2), d) size 1x1, 2x2 and 3x3 (MaxSquareSize = 3)*

[Figure]

*Same as figure 9 (left only) with decomposition only with a) square clusters of size 1x1, b) size 1x1 and 2x2 (MaxSquareSize = 2), c) size 1x1, 2x2 and 3x3 (MaxSquareSize = 3).*

Here, it is clear that when using larger structures, the L distribution shifts to higher values while P takes on lower values. Conversely, decomposing with either 1x1 or 1x1 and 2x2 structures from a scene that contains larger ones creates the impression that the organization is less random, as P reaches maximum values. This supports the fact that our approach is best suited to more random characterization when using dominant shapes within the scene.

It is acknowledged that more sophisticated decompositions could be proposed. However, it is asserted that the present decomposition is adequate, given its limited computational complexity and its capacity to discriminate adequately in the generated (empirical) distribution of L. This preliminary approach may be superseded in further work.

Additionally, would one expect such regularly shaped (e.g., circular or square) deep convective cores in environments with strong vertical wind shear? In such cases, convective elements may be elongated or tilted, which may not align with the chosen geometric representation. A short comment on the sensitivity of the method to these physical variations would be valuable.

The circular assumption relies on a vertically erected plume model which is a useful idealization of deep convection dynamics. In this simple case, the dynamics is indeed circular. Yet if the plume is slanted (due to wind shear), the cross section of the plume would look like an ellipse. The circular assumption (which translates into square pixels) should be understood as an idealized, first order approximation. To better convey this dimension of our work, the title has been modified.

Figure 4: "larger structures are associated with stronger and deeper convection"

It might be useful to mention that this relationship is consistent with observations. For instance, Moseley et al. (2019) show that larger convective cells tend to exhibit more intense precipitation.

Yes, it is also consistent with radar observations, starting with the GATE campaign (Betts and Houze, 1981). Note that Moseley et al. reports simulations-based results.

L. 354 355: "characteristics of the scene"

Consider referring to the "Scene characterization" section below to precise the variables that are retained for it.

OK done.

Figure 6:

"length" → "length"

Yes done.

Some text are in bold font or in italics without obvious reason to me.

Consider adding a yellow and an orange line for step 5 as these processes are also applied to the random grids.

Yes, thank you, we've updated the incriminated figure as follows:

[Figure]

L 367: Scene area

Could the authors clarify how the scene area is precisely defined in the analysis? From Figure 3, it does not appear to correspond to the minimal rectangular bounding box enclosing the cloud shield.

Yes, indeed, the definition of the rectangular outline can be specified in more detail. It is the rectangle that most closely frames the outline of the cloudshield, but within the geometry of the TRMM/GPM swath, without the possibility of extending beyond it because the radar

information is missing there. We therefore retain a rectangle oriented in the direction of the satellite's orbit, with a margin of 1 pixel at the edges.

It is acknowledged that the TOOCAN-radar dataset is subject to certain biases, primarily concerning the delineation of this rectangular contour. Additionally, it has been observed that certain systems are not fully encompassed within the extent of the radar swath. However, we have limited this as much as possible by only retaining clouds for which at least 70% of their surface area is within the swath. It should be noted that these biases are statistically offset by the large number of cases contained within the database.

Since the scene area directly affects both the computed convective fraction and the generation of the reference random distribution for the spatial organization metric (P), its definition is critical.

Regarding the influence of the grid size chosen on the four variables, S is an absolute measure and L does not depend directly on the size of the rectangular contour, only on the number of pixels and their arrangement in the grid. On the other hand, for F, it is true that we introduce a more or less constant bias, tending to minimize F in comparison with the convective fraction with respect to the cloudshield (which is different). Here, it is the convective fraction in the rectangular scene, that is an algorithmic variable approximating the physical convective fraction. This is indeed a possible refinement for a future version that will need to be considered, but we show in the rest of the answer that F is not too sensitive in the final discrimination process when keeping <5-10% of error (see figures B3 answering a comment below, assessing the sensitivity to the four parameters in the classification), and F is kept constant for the generation of random scenes (same bias).

In particular, I am concerned about potential biases introduced by the shape of the cloud shield. For instance, in the case of an elongated DCS, the encompassing scene area may include large regions outside the actual cloud shield — areas that do not contain any DCCs. This could lead to an artificially low convective fraction and/or a misleadingly high organization score (e.g., P close to 1), depending if the DCS itself is densely populated with DCCs.Conversely, for more rectangular DCSs, the scene area might better reflect the actual cloud shield, resulting in more representative values of these metrics.

Regarding the variable P, it is true that the current process can artificially push P towards higher values, since random pixels are allowed throughout the rectangle and not within a more rigorous cloud mask. This will also be the subject of potential improvements in the future, even though we show here that the sensitivity of these choices does not require a revision of the major results of this study (see additional sensitivity figures from Monte-Carlo estimates). Thanks a lot for pointing this out.

The DCS shown in Figure 14d is a good example of the limitations of the current definition of convective fraction based on the rectangular scene area. Although this system appears to have a high proportion of convective precipitation pixels relative to stratiform ones (possibly >50%), its computed convective fraction is only 7%. In contrast, the DCS in Figure 14c has a visibly lower ratio of convective to stratiform pixels, yet its convective fraction is twice as high. This discrepancy appears to stem from differences in cloud shield shape — with case c being more rectangular and thus more tightly filling the bounding box used as the scene area.

Here again, it is necessary to differentiate between the convective, stratiform precipitating part of the cloud (which is physical) and the convective fraction, which represents only the number of convective pixels within the rectangular grid (as a variable used for the algorithm described in this work). Since we have demonstrated that F could incur an error factor of 5–10% in the classification process without affecting the overall distribution of the classes, we believe that this approximation is sufficient without the need for more sophisticated methods (cloud masks, refined contouring, etc.).

This illustrates that the current definition of F is highly sensitive to cloud shield geometry, particularly for elongated or irregular systems, and may not reflect the true convective content of the DCS. I suggest that the authors more explicitly discuss this limitation and consider whether an alternative scene definition — e.g., based on the actual cloud shield contour, the area of precipitation, or a convex hull — might reduce this bias.

In conclusion, we are aware of this limitation, which is inherent in the diversity of cloud shield shapes observed in this satellite database. Indeed, there are several orders of magnitude in size, and clouds are sometimes elongated, or have parts outside the swath of the TRMM/GPM satellite.

F is not a direct physical variable and remains difficult to estimate visually because scale and edge effects must be taken into account depending on the size of the cloud. However, we have opted for a rectangular outline in the geometry of the radar swath for the sake of efficiency and generalization in the calculations for the subsequent steps in the algorithm, assuming that edge or distortion effects are not predominant. It should be noted that most clouds are on average fairly circular, sometimes slightly elongated, but rarely with a width-to-length ratio greater than 3 (see figure below that illustrates the distribution of the eccentricity (length of the minor axis to the length of the major axis) of the >60 000 cloudshields within the TOOCAN-radar dataset.

[Figure]

*Empirical probability density function (estimated via KDE) of the* eccentricity (length of the minor axis to the length of the major axis) of all the sampled cloudshields within the TOOCAN-radar dataset

It should also be noted that these limitations are much less pronounced for the TOOCAN-RCE dataset, which has rounder and less regularly elongated clouds, and which has no swath or grid geometry limitations. The consistency of the results between the TOOCAN-radar and

TOOCAN-RCE datasets shows that for this study, our approximation is sufficient, although refinements are possible for future studies.

L. 394: "convex"

The use of a "convex" contour to define the central region of the autocorrelation field is somewhat unclear. In Figure 7, the contour shown does not appear strictly convex or minimal. Could the authors clarify how this contour is derived and to what extent its geometry affects the estimation of the typical aggregation distance?

I am concerned that applying a convex hull to potentially irregular or elongated shapes may artificially increase the value of L. Since L is defined as the maximum internal distance within the convex hull — rather than within the original (possibly non-convex) shape — the metric may become sensitive to shape distortions, particularly for highly anisotropic or fragmented patterns. This could introduce a systematic overestimation of L for some systems but not others, depending on the complexity of the spectral power field. I suggest that the authors clarify how often such distortions occur in practice, and whether they have assessed the sensitivity of L to the convex hull approximation.

Firstly, it represents convexity at pixel level, so it cannot be perfectly convex. This precaution was originally included to ensure that L could be approximated by the major axis of an ellipse encompassing the 10% contour. This illustrates that we are looking for a characteristic length within a simple shape. The value of L is not modified by the convex contour; it is only modified by choosing the central contour from the threshold mask T, and all sensitivity depends on T. This has been addressed in the paper. This is an over-engineered process that could possibly be removed in future versions, as it is useful only in very rare cases and does not modify the estimation of L or the classification process at all.

Fig. 7:

Consider representing the metric L on this figure.

Done with the green arrows on Figure 7 (+ updated caption).

L. 408: "anisotropy"

The mention of anisotropy feels somewhat out of place, given that it is not used in the method. That said, it raises an interesting point. One could imagine that a variable quantifying the anisotropy of the DCC distribution might complement the current set of metrics used to characterize spatial organization. While I understand the motivation to limit the number of variables for clarity and robustness, this example illustrates that additional descriptors could be considered in future work for a more refined or context-specific characterization of convective organization.

Yes, we are also interested in this notion. It should be noted here that part of this anisotropy is taken into account in the autocorrelation by definition and by extension in L, but that it is indeed a potential refinement that would require decorrelating the anisotropy factor from the other components of the autocorrelation 2D. This will be potentially included in future work as it does not align with our primary objectives for the presented paper.

L. 415-416:"spatial morphology" vs. "condensed structural information"

I find the distinction between "spatial morphology of the raw field" and "condensed structural information captured" somewhat unclear. Could the authors clarify what is meant by this difference?

Yes, we agree this needs to be reformulated. We wanted to illustrate that L is a measure of physical distance as well as condensed information about the spatial arrangement and periodicity.

"As a result, L encodes both the spatial morphology of the raw field and the condensed structural information captured by the autocorrelation"

⇒ "As a result, L encodes both the perception of the spatial extent of the main structures of the raw field and the condensed structural information regarding the spatial arrangement of these structures captured by the autocorrelation."

L. 420:

What is "C"?

Replaced by F.

L. 424, 427, 459, and elsewhere: Use of "probability" to describe P

In several places, P is referred to as the "probability of the scene being randomly organized" or "probability of deviation from a random distribution." However, as I understand it, P is more precisely defined as the percentile rank of the scene's characteristic length (L) within a reference distribution derived from randomly generated (uniform) DCC patterns. It does not represent a statistical probability in the formal sense.

For clarity and consistency, I suggest revising the terminology throughout the manuscript to reflect this. Referring to P as e.g. a percentile-based measure of deviation from randomness would more accurately describe its meaning and avoid potential confusion with probabilistic frameworks.

Yes, indeed, all the occurrences of "probability" have been modified or contextualized throughout the text as suggested by the referee.

L. 431 and elsewhere: "bootstrapping"

As I understand it, the authors generate randomized spatial patterns of DCCs by redistributing a fixed number of DCCs uniformly within the scene. Since this process does not involve resampling from the observed data with replacement, it does not correspond to a formal "bootstrapping" procedure. Rather, it is more accurately described as a Monte Carlo approach for generating a reference distribution under spatial randomness.

I recommend updating the terminology accordingly to avoid confusion with statistical bootstrapping methods.

Again, all the occurrences of "bootstrap" or equivalent have been modified throughout the text to meet the referee's suggestion.

L. 443: "500"

I wonder whether 500 samples are sufficient to robustly estimate the distribution of L under the assumption of a uniform DCC distribution. While some sensitivity analysis is provided in the appendix, the figures are somewhat limited in really showing the accuracy or convergence of the Monte Carlo approximation in each case.

Since P is a percentile rank estimated from a finite sample, its uncertainty can be approximated as $\sqrt{P(1-P)/n}$, which implies a standard error of about ±1–2% for n = 500. This level of uncertainty is likely acceptable for the broad classification presented in this manuscript.

However, it may become problematic if users wish to compare two scenes with similar P values. I suggest the authors consider including a more detailed justification of this sample size or provide confidence bounds on P where relevant.

This point is partially addressed in Figure B2 in the Annex section. As demonstrated by Figure B2 (where, even for 100 realizations, the distributions' parameters are highly comparable to the one with 500 realizations) and the numerous experiments conducted during the course of this study (not shown), 500 realizations are a satisfactory compromise between computational time and the quality of the generated L distribution across the two datasets. The estimation of P is highly consistent regarding the M factor, be it 100 or 1000.

Moreover, we also carried out an integrated sensitivity study, the details of which are provided below, at the end of this document. This study was conducted to address the P sensitivity; see Annex B3 (added). Please see also the detailed response to one of the final comments.

L. 447: "spatial organization" → "spatial aggregation"?

Done ⇒ "spatial arrangement"

L. 465: "such a specific spatial arrangement (or its equivalent) almost never occurs in the randomly generated scenes"

This statement could be made more precise. A scene being rare under the assumption of randomness does not, in itself, indicate whether the spatial distribution is clustered or regular. Since the main goal is to assess clustering or aggregation of DCCs, I suggest rephrasing this to highlight that P captures deviation from randomness, and that high P values specifically indicate increased clustering, rather than rarity alone.

We don't use the clustering notion to illustrate the meaning of the value of P. Besides, rarity is somehow a vague notion that can lead to ambiguous interpretations. We don't mention it directly. Instead, we express the notion of how many times an event occurs within the total number of attempts in the context of the generated scenes, as the percentile computation. Besides, as described in the paper, our objective is broader than simply distinguishing between aggregated and clustered, as with metrics such as Iorg/Lorg, for example. Rather,

we aim to quantify the deviation from randomness and the structuring of convective cores. Therefore, we believe that the original phrase is appropriate in this context.

L. 472-473: "… the generation of the ensemble of scenes with a given convective fraction (F) is constrained by a few parameters of the stochastic model …"

I would like to point out that the scene area is also a critical parameter influencing the metrics (see my comment above on scene area).

Modified as follows ⇒ "Indeed, the generation of the ensemble of scenes given a convective fraction (F) and convective area (A) is constrained …"

L. 475-477: "sensitivity of F"

The current analysis of the sensitivity of the L distribution to the convective fraction F (Figure 10, B1, B2) is informative but could be presented in a more visual or intuitive way. For example, plotting histograms or kernel density estimates of L across several discrete values of F could help the reader more clearly understand how F shapes the reference distribution and, by extension, the behavior of P. This might be more accessible than the current presentation.

Thank you for this feedback. The current representation aims above all to show that the evolution is non-linear as a function of F. Furthermore, the distribution of L is often almost symmetrical or pseudo-Gaussian (see figure below that is the L distribution for several F values, with the mean value represented in red), regardless of the value of F, but has activation thresholds that depend on the dominant second-order structures. We consider our original figure to be more informative and appropriate, as the main message is more about the activation of unconstrained data ranges for certain values of F and not the centering of the distribution in L (already visible with the blue shaded areas representing the +/- 1$\sigma$.

[Figure]

*Empirical probability density function (estimated via KDE) of the generated distribution of characteristic length L for several F values, with the same condition of experiment as in Figure 10, right. Average value is shown in red.*

L. 484: "challenging to compute a meaningful probability"

The phrase "meaningful probability" is somewhat vague, and it's not clear from Figure 10 what specifically supports this statement. If the issue is uncertainty in L and P for extreme

values of F, it would strengthen the analysis to move beyond the empirical filtering approach (e.g., excluding scenes with F < 8% or F > 25%).

Instead, I suggest a more quantitative uncertainty-based filtering method. For instance, the authors could:

1. Estimate or infer the uncertainty in L,
2. Combine this with the Monte Carlo sampling error (as discussed above) to deduce a total uncertainty in P,
3. And exclude scenes for which the total uncertainty in P exceeds a given threshold (e.g., ±5%).

This would provide a clearer and more defensible basis for filtering scenes, and would strengthen the methodological transparency of the approach.

Again please find the detailed answer at the end for the final comment that addresses those points in detail, completing the original sensitivity study.

Figure 10, B1, B2:

Please indicate what the shaded area represents.

OK added in the caption.

L. 499: "grid size" → "scene size"?

OK done.

L. 518: "consistent with the unfiltered distribution"

Please indicate whether this is shown in the paper or not.

OK done ("not shown" added).

L. 534:

See comment for line 465.

Again the sentence in question appears to be correct from our perspective. See the answer given for the precedent comment.

L. 535: "distribution" → "distributions"

OK done.

L. 549: "Empircal metrics … show"

The wording here is slightly misleading. The metrics are quantitative and not empirical in nature. It is rather the interpretation of these metrics that involves empirical reasoning or subjectivity.

OK done ⇒ "Classical clustering metrics".

L. 551: "Figure 12" → "Figure 13"?

OK done.

L. 559: "more randomly distributed" → "less aggregated"

Modified ⇒ "less clustered".

L. 562: "This indicates that F alone is not a discriminating variable …"

I do not find that the last two sentences indicate that F is not a discriminating variable (but maybe more the end of this sentence), please reformulate.

⇒ L562 : "This indicates that F alone is not sufficient to fully discriminate between organization types, as the spatial arrangement strongly influences the other three key parameters."

L. 563: "very unlikely to be randomly spatially distributed"

This could also be the case of regular patterns in the other end of the distribution. Please specify.

OK modified ⇒ "very unlikely to be randomly spatially distributed with high L values"

L. 564: "do not necessarily have high F values"

The statement that "highly organized scenes do not necessarily have high F values" is phrased as if it were surprising. If this is indeed counterintuitive or contrasts with previous assumptions, it would be helpful for the authors to provide citations or context to support why this is the case. Otherwise, I suggest rephrasing the sentence to avoid implying that this decoupling is unexpected.

OK modified removing "necessarily".

L. 566: "(Figure 12)" →"(Figure 13)"?

OK thank you, we modified it.

L. 578: "Figure 11" → "Figure 13"?

OK thank you, we modified it.

L. 546-585:

I find it difficult to form a synthetic view of the organizational classes after reading this paragraph, as the presentation feels somewhat interwoven and comparative from the start. I suggest restructuring the paragraph to improve clarity: the authors could first systematically describe the main characteristics of each class individually, and only after that proceed to

comparisons across classes. This would help readers better understand and differentiate the classes before being asked to contrast them.

The classification from random to organized is based on the P distribution (and secondary on the A and L variables for which the order is the same in terms of means and modes of distribution), that is more and more close to 1 as the class number increases. Note that class 0 and class 1 are not much closer to each other than any other Class.

We do not see the point in refining the details on each class individually before defining them as a physically sound family of convective patterns (which is done at the end of the paragraph), considering that in such clustering process, each class is distinguished by its major differences from the others, as detailed in the original text.

Nevertheless, we included a modification in the dedicated paragraph to complete the original description before comparing the classes: "As P increases (and secondary L and A too), so does the level of organization. The order of the classes follows this progression, starting with class 0 as the least organized and ending with class 3 as the most structured."

L. 621-631:

I think it is also important and interesting that the authors compare their percentile-based organization metric (P) to the organization index Lorg introduced by Biagioli and Tompkins (2023). While Lorg is based on nearest-neighbor distances and does not require scene-level resampling, P is derived from the maximum spatial extent of the autocorrelation field and relies on Monte Carlo sampling.

It would be helpful if the authors could briefly discuss how these differing foundations may affect the quantification of the spatial aggregation of DCC and under what conditions one metric might outperform or complement the other.

Yes, thank you for the suggestion. Extending our comparison to other existing metrics would probably complement the work already carried out in the Discussion with COP, ABCOP and ROME metrics. It is important to note though that Iorg and Lorg are point process-based metrics differ from the present object-oriented approach. Iorg/Lorg offer a completely different way of quantifying *organized* convection, as they do not account for the areas of the convective objects involved, only their centers (e.g. their number). Actually, both these methods focus on the *clusteriness* of the center of the convective cores as explained in the Introduction.

Furthermore, Iorg is notably sensitive to the number N of objects (with a high negative correlation between Iorg and N) and their relative distances. This necessitates a stable number of objects (greater than 35) for reliable measurements. (Mandorli et al., 2024; Semi & Bony, 2020), which is often not possible in our data, even when using our decomposition process.

In summary, reconciling and comparing the P and Iorg/Lorg metrics may not prove easy and possibly not even interpretable due to the strong differing assumptions between the two methods. In particular we are not suggesting that P should be used as a standalone, continuous metric like Iorg, so the comparison may not be very meaningful. In any case, such a study is beyond the scope of this paper.

L. 642: "However, the fact that ABCOP distinguishes the classes more effectively than ROME suggests that while organization is more influenced by the total convective area (S), it cannot be fully captured by this single variable or its combination with F".

I may be misunderstanding the logic here, but the sentence seems to suggest that because ABCOP outperforms ROME, this implies that spatial organization cannot be fully captured by S combined with F. It would be helpful for the authors to clarify the reasoning behind it.

OK modified ⇒ "However, the fact that ABCOP distinguishes the classes more effectively than ROME suggests that organization is more influenced by the total convective area (A) than the mean convective object area. Nevertheless, the overlaps within ABCOP distributions illustrate that the complexity of spatial arrangements cannot be fully captured by only A or its combination with F."

L. 657: "In this study, a new method is introduced"

This sentence would benefit from a clearer link to the utility or purpose of the method, as it currently appears disconnected from the previous discussion. I suggest rephrasing to briefly restate what the method is intended to achieve.

OK modified ⇒ "In this study, a new method is introduced to quantify and characterize spatial arrangement of convective areas within a specific grid encompassing the instantaneous cloudshield of a deep convective system (DCS)."

L. 663: "establishing a probabilistic distance from a random spatial organization"

Not really (see previous comments). Please also note that the Lorg index (Biagioli and Tomkins, 2023) is also a measure of the deviation from a random spatial organization.

Here, the referee refers to the work of Biagioli and Tomkins, 2023, but to our knowledge, it is the OII index that measures this deviation, not Lorg directly (see citations below). Furthermore, Lorg is used in point-processes, not in object-oriented processes, unlike our approach.

> "To summarize the departure from randomness, we introduce a second index, the organization irregularity index (OII)" (Biagioli and Tomkins, 2023)

> "the organization irregularity index (OII), which is an integrated measure of the departure of convection from randomness across all spatial scales. Thus, Lorg and lorg give two integrated assessments of the mean organization, while the OII is an integrated measure of the variance of organization." (Biagioli and Tomkins, 2023)

However, we have reformulated as follow: "establishing a probabilistic measure of the deviation from a random spatial arrangement"

L. 700:

Missing closing parenthesis.

OK done: "[...] for instance)."

Conclusion:

As I understand it, the TOOCAN algorithm defines Deep Convective Systems (DCSs) such that only one convective seed is permitted within each cloud shield. Given that deep convective cores (DCCs) typically coincide with the coldest cloud-top temperatures, this definition could influence the resulting spatial organization of DCCs within individual DCSs. In particular, constraining the number of convective seeds may limit the diversity of spatial arrangements that the method can capture.

We thank the referee for this remark. The TOOCAN algorithm does indeed identify convective seeds, but this procedure is based on a region-growing technique implying an iterative process of detection of convective seeds at several thresholds followed by a watershedding procedure applied on a spatio-temporal volume of infrared brightness temperature. Deep convective systems (DCSs) and their associated cloud shield are then identified throughout their full life cycle. However, these IR-based DCS definitions are not directly comparable to the deep convective cores (DCCs) detected in our study using spaceborne radar, which provides direct information on the convective precipitation cores rather than the extent of the cloud shield. The sensitivity of the TOOCAN seed definition has been carefully evaluated in dedicated publications, and it does not affect the collocation procedure applied here. More importantly, it has no impact on the identification of DCCs in the radar-derived precipitation fields used in this work.

While this does not undermine the value of the proposed approach, I suggest briefly discussing this methodological dependency in the conclusions—especially regarding the sensitivity of the characterization framework to the specific MCS tracking algorithm used.

The TOOCAN algorithm has been used in intercomparison exercises (Prein et al., 2024; Feng et al., 2025) and shown to perform well in identifying the life cycle of MCS. Yet here we are not dealing with MCS only but with the full spectrum of DCS for which such an intercomparison does not exist. Instead TOOCAN has been used in many physical studies showing its good performance (e.g., Elsaesser et al., 2022 among others) revealing its ability to depict a physically sound cloud shield life cycle for the wide spectrum of DCS. This gives us some confidence in using TOOCAN for this purpose. This said, our results are only valid for TOOCAN segmentation and only relevant in this specific context. We hope we have described in enough detail our technique that any interested readers shall be able to implement it for any other tracking algorithms and assess the relevance of the results in that case.

L. 888-889: "a sensitivity analysis revealed that this non-uniqueness has no significant effect on the subsequent methodology"

Please indicate that this is not shown in this paper.

OK done.

L. 905: "Figure B2" → "Figure B1"?

Yes thank you, we modified it.

L. 909-912: "the most suitable","reasonable"

The use of terms like "most suitable" and "reasonable" in reference to scene selection feels vague, especially when based solely on empirical observation.

As mentioned earlier, it would strengthen the methodological rigor of the study to define selection criteria based on a quantitative threshold — for instance, by estimating the uncertainty on P for each scene and excluding those with uncertainty above a predefined limit. This would provide a more transparent and reproducible basis for filtering.

To answer this comment and some others that came before it, we present here several elements that complete the sensitivity study that has been already carried out in the manuscript:

A Monte Carlo-style study was conducted on K-means clustering, adding noise to each parameter in the radar dataset (~60,000 points). For each parameter, we added uniform noise with amplitude [-alpha/100*(Perc_90-Perc_10),+alpha/100*(Perc_90-Perc_10)], with alpha in % for values [0.5, 1, 2, 5, 10, 15, 20, 30]. Then we reclassify using the same Kmean algorithm (under the same conditions), and compare the class labels, calculating the number of points that change classes (or their proportion), the change in the silhouette score, and the Adjusted Rand Index (ARI), which quantifies the agreement between two partitions, correcting for random fluctuation. An ARI close to 1 means a very high correspondence between the two partitions, and when it approaches 0, we are close to an agreement corresponding to random draws. We perform this process 10 times for each value of alpha to estimate uncertainty on the mean values of each of the metrics, for each of the parameters. The results are presented in the figure below (that corresponds to the added Figure B3).

[Figure]

We summarized here several key findings that are described in the dedicated section in Annex B:

⇒ The order of importance regarding parameter sensitivity in classification is F, L, P, S

⇒ Up to +/- 5% noise on all parameters, the classification remains stable (with ARI>0.9, and even up to >10% for S, P, and L. Only F shows greater sensitivity above 10%

⇒ The uncertainty estimated with 10 experiments for each of the selected set of parameters/alpha combination also shows that the method is statistically robust.

We do not have the complete error model with error propagation in L and P (as it is difficult to access), we acknowledge that. However, thanks to these experiments, we have the consequences of a possible systematic disturbance/error on one or other of the parameters, and we note that within a range of approximately +/- 5-10% error, which is relatively significant, the classification is very stable and therefore considered robust.

For example, for 500 realizations, using the formula given by the referee on the uncertainty of P: $\sqrt{P(1-P)/n}$, we have a maximum error of approximately 2% (for values of P close to 0.5, but this drops to around 1% for values >0.9 (which is the majority of cases in this dataset), so it has no impact on the classification (see figures a, b, c of Figure B3).

L. 914: "We prove (not shown) that it would be interesting …"

Please reformulate.

Modified as follows: "We suggest that it could be interesting in such cases to increase the contour threshold up to 15% to better capture a proper range for L values."

L. 924: "histograms" Figure B1 and B2 do not show histograms (... although it would be helpful to see histograms here, see comment on L. 475–477).

⇒ "histograms" replaced by "distributions"

Figures B1, B2: "Same as figure 9 ..." → "Same as figure 10"

⇒ modified in figures' caption

---

## Referee Report (RR1)

I appreciate the authors' efforts in addressing my previous comments and revising the manuscript based on the initial feedback. While most of the authors' responses satisfactorily address my concerns, some aspects regarding the robustness of the methodology remain questionable, despite the additional sensitivity analysis provided.
In this analysis, the authors examine how much the classification changes when noise is applied to four parameters — the convective fraction (F), convective area (A), characteristic length (L), and percentile (P). This is a valuable addition, as the manuscript presents a methodological approach. However, the analysis only assesses the method's sensitivity to errors in these parameters, without evaluating the typical magnitude of such errors in practice.
Based on the authors' response regarding the definition of the scene area, I am concerned that typical errors in F (and possibly P) could exceed the 10% threshold mentioned in the sensitivity analysis. I encourage the authors to either provide evidence that this is not the case or revise their methodology for defining the scene area. I also have follow-up (more minor) comments regarding the interpretation of P and the sensitivity to the tracking algorithm. Please see my detailed comments below.

Scene area

I thank the authors for their detailed explanation of how the scene area is defined for each case. To my understanding, the scene area corresponds to the smallest rectangle that fully encloses the DCS (with one pixel of padding) and is aligned with the radar swath grid. I see two potential sources of bias in this definition: (1) the shape of the DCS, particularly its deviation from a rectangular form, and (2) its orientation relative to the grid.

Non-rectangular DCSs that are oriented at approximately 45° relative to the satellite grid will result in significantly larger scene areas than those that are rectangular and aligned with the grid. For the method to yield a meaningful classification of convective element arrangements within DCSs, such orientation- and shape-related biases are undesirable. While it is acceptable for F to be treated as an algorithmic variable representing the convective fraction, this remains valid only if there is a clear and consistent relationship between F and the actual convective fraction (i.e., larger F values should correspond to higher convective fractions). Based on my understanding of the current methodology, I am concerned that this relationship may not hold given the current definition of scene area. To illustrate this point, consider a simplified example of an elliptical DCS (aspect ratio ≈ 1/3) observed by two radars with different grid orientations (cases A and B in the figure below). As it is the same DCS, the method should ideally yield similar convective fractions and classifications in both cases. However, as shown, case B produces a significantly larger scene area than case A — a difference of approximately (30–22)/22 = 36%. This increase in scene area could lead to a decrease in the derived convective fraction by about 26%, potentially resulting in different classifications for the same physical system depending solely on its orientation relative to the radar grid (as suggested by the additional sensitivity analysis).

Differences in DCS geometry would add on this orientation bias, making the convective fraction F less representative of the actual physical convective fraction. Large variations in scene area would also influence the P metric, which is likely to be larger for case B than case A, possibly exceeding the 15% threshold reported in the sensitivity analysis.
I encourage the authors to either (1) provide evidence that typical errors in F and P do not exceed the thresholds discussed in Figure B3, or (2) revise their method to reduce these biases. In addition, I recommend that the authors include a clear description of how the scene area is defined in the manuscript, as this aspect is critical to the methodology and

may substantially influence the classification results. Finally, the limitations associated with the definition of scene area should be discussed in the "Limits of applicability" section.

[Figure]

Figures:

Some figures still lack panel labels (letters) for reference. This is particularly the case for Figure 14, where letters are mentioned in the caption but do not appear in the figure itself. For clarity and consistency, I recommend adding panel letters (e.g., a, b, c, …) to all relevant figures, as these are preferable to references such as "left" and "right."

Interpretation of the percentile P:

- L. 465-466: "computing a percentile P, which quantifies the deviation of the real scene's spatial arrangement from a random arrangement (Figure 9, Red arrow in Figure 6)."

That wording implicitly assumes that higher percentiles mean stronger deviations from random.
But that is not necessarily true, low percentiles can also represent strong but opposite-direction deviations (more regular patterns instead of random). Please reformulate.

- L. 470-473:

For clarity, I copy the Authors' response to my comment on the previous round of revision.

"We don't use the clustering notion to illustrate the meaning of the value of P. Besides, rarity is somehow a vague notion that can lead to ambiguous interpretations. We don't mention it directly. Instead, we express the notion of how many times an event occurs within the total number of attempts in the context of the generated scenes, as the percentile computation. Besides, as described in the paper, our objective is broader than

simply distinguishing between aggregated and clustered, as with metrics such as lorg/Lorg, for example. Rather,we aim to quantify the deviation from randomness and the structuring of convective cores. Therefore, we believe that the original phrase is appropriate in this context."

I understand that the objective of the paper is broader than simply distinguishing between aggregated and clustered patterns, and the proposed method indeed achieves this, as indicated by the combination of P with the three other variables. However, my comment focuses specifically on the variable P.

The variable P measures the fraction of randomly generated scenes that have a lower L than the actual scene. Therefore to my understanding, the variable P, as defined in the current version of the manuscript, is more of a measure of the degree of clustering expressed in a statistical sense (on a probabilistic scale) than a measure of the deviation from randomness. This is because both very low and very high values of P indicate strong deviation from randomness, as noted above. Conversely, as P increases from 0 to 1, the spatial arrangement transitions progressively from more regular to more clustered scenes. If the Authors would like to put more emphasis on the deviation from randomness than the degree of aggregation, I would suggest to use a modified metric, for example $1-2.\min(P,1-P)$, that will more closely quantify this aspect.

- L.539-540 "On the other hand, the P distribution, indicating the probability of the characteristic length to differ from a random arrangement (Figure 12, d) is very much skewed to the highest values (around 1) with a very long tail. This indicates that only very few scenes can be associated with a randomly generated pattern."

The P distribution does not directly indicate a probability of differing from a random arrangement. First, very low P values are also indicative of deviation from randomness, as noted above. Second, such P-values (percentiles) only quantify the conditional probability of observing a value less extreme than the observed statistic (here L) under the null hypothesis (here, H0 = randomness (H1 = non-randomness)). Formally, they measure $\text{Prob}(L \le L_{obs} \mid H0)$, which has no direct relation with $P(H1 \mid L \le L_{obs}) = 1 - \text{Prob}(H0 \mid L \le L_{obs})$.

To illustrate, consider a bag containing many dice — some fair, some modified. You randomly draw one die, roll it, and obtain a six. Under the null hypothesis of a fair die, the probability of rolling a six is 1/6, and thus the P-value is relatively high (not indicative of deviation). However, this outcome alone tells us nothing about whether the die was fair or not. The P-value only quantifies how likely the observation is if the die were fair; it does not give the probability that the die is unfair given the observation.

Similarly, in your analysis, the P metric does not directly represent the probability of non-randomness. While the mapping between P and the probability of deviation from randomness may appear more meaningful when the reference (random) distribution is approximately bell-shaped, there is no one-to-one correspondence. The interpretation of P thus depends both on the shape of the null distribution and, in a Bayesian sense, on the prior probability of non-randomness, which can vary across cases.

- L. 574: "very unlikely to be randomly spatially distributed with high L values"

This statement remains somewhat ambiguous. It could be misinterpreted as referring to the unlikelihood of observing L relative to all high values in a random distribution, rather

than the intended meaning: that the observed spatial pattern is specifically unlikely under the null hypothesis and falls on the side of clustered rather than regular patterns.

Sensitivity to the TOOCAN seed definition:

In the previous round of review, I shared my concern that the method used for detecting DCSs in the TOOCAN algorithm may restrain the diversity of spatial arrangements of DCCs, even when these DCCs are identified through radar measurements.

While the response provides a useful description of the TOOCAN procedure, it does not demonstrate that the IR-based identification of convective systems is fully independent of radar-based DCC detection (microwave measurements). Given that cloud-top temperature and convective precipitation intensity are often correlated (e.g., Arkin and Meissner, 1986), it seems surprising to me that the TOOCAN definition of DCS, which emerge from single convective seeds, would have no impact on the spatial arrangement of DCCs identified in radar.
Since convective seeds in the TOOCAN algorithm are defined as very cold brightness temperature minima, and only one seed is retained per DCS, one would expect the radar-identified DCCs to be spatially close to these IR-based seeds. This makes it difficult to conceive how the TOOCAN DCS definition could have no influence on the spatial distribution of radar-detected DCCs. A brief quantitative check (e.g., distance statistics between TOOCAN seeds and radar DCC centroids), or reference to an earlier validation study would help clarify this point.

Reference:
Arkin, P. A., and B. N. Meisner, 1987: The Relationship between Large-Scale Convective Rainfall and Cold Cloud over the Western Hemisphere during 1982-84. *Mon. Wea. Rev.*, **115**, 51–74, https://doi.org/10.1175/1520-0493(1987)115<0051:TRBLSC>2.0.CO;2.

---

## Author Response (AR2)

Dear authors,

Thank you for your revised manuscript. I have obtained one further review from one of the initial referees. From that and my own reading, we feel that you have addressed most of the points well, but the referee has some remaining concerns - particularly, about how the definition of scene area and orientation of a feature could lead to larger than expected errors. I am not sure if we have misunderstood something or whether the orientation with respect to the radar is indeed a potentially large issue. I have attached these comments (see pdf) and would appreciate if you would be able to respond to them and modify the manuscript as you feel appropriate. Other than these, I feel and the reviewer agrees that comments have been addressed well. Please let me know if you have any questions!

Best wishes,

Andrew

Dear Editor

Before we answer your and the reviewer questions, we would like to emphasize a few points. First of all, the first round of review provided a very interesting in-depth set of questions thanks to the appreciated efforts of two reviewers. The fact that, now, you think that there is a "potentially large issue" with our work is puzzling. We are surprised that you decided to call for major revisions after a round of long, yet minor revisions. The reviewer is using words like "critical", "substantial", etc. which we think are not relevant to the discussion at this stage.

First it is important to note that our work is not a house of cards; it will not collapse just because of one aspect. The only potential issue is the delineation of the scene area and the orientation of the shield when using space borne radar.

As emphasized in the original manuscript, the method does not only apply for such narrow swath scanning radars. Half of the analysis provided in the manuscript indeed concerns outputs of model simulations free from radar geometry. Our technique can be applied to various other satellite borne instruments, for instance from passive microwave observations of hydrometeors for which the swath width is much larger than the radar one as discussed for decades in the literature (e.g., Nesbit et al., 2003). So, if the geometry were an issue, it would not be a "*critical"* one for using the method. At best, it could be a limitation of the technique for these particular radar observations.

Also, the consistency between the results from the radar-based analysis or the model-based outputs rules out any "*large issue*" with the method.

The source of uncertainty pointed out by the reviewer can arise from specific alignment of the radar swath together with a specific aspect ratio of the convective cloud shield. As shown in our responses during the previous review round, the distribution of aspect ratio of the cloud shield is smoothly distributed across the range of possible values and is centered at 0.5. So only ~50% of the cases can qualify for alignment computations, and

even fewer if very elongated systems are concerned. Therefore, in any case, this potential source of uncertainty cannot again be "*large*".

This said, it is interesting to identify such a source of uncertainty and we thank the reviewer for pointing it out to us. The reviewer suggests that our estimations of F (and possibly of P) could be unrelated to the actual convective fraction and that such an uncertainty can impact our classification. In the following, we provide new computations confirming:

1) our estimates of the scene area are very consistent with the true cloud shield area
2) our estimates of F are very consistent with the true convective fraction
3) the uncertainty on the estimation of F does not impact significantly our classification
4) we also discuss the possible impact on estimating P and finally address the comments on the TOOCAN algorithm.

We think it is interesting to specify these few cases to warn the reader of a cautionary interpretation of the computation of F and P in some cases of the radar configuration. As suggested by the reviewer we have added a dedicated paragraph in the section "Limits of applicability" that reads:

L483: "In the case of the narrow satellite observations of the radar, the estimation of the shield area (and, by consequence, the F parameter) using an along track aligned rectangle yields a biased estimation of this area. Nevertheless, this 'algorithmic' variable correlates very strongly with the true area and does not overall impact our methodology. But, in the case of elongated systems (eccentricity < 0.3) with a specific alignment with respect to the along track, in the range of 30° to 45°, this effect could be significant. We underscore the need to use our technique with caution for these infrequent cases: combination of cloud shield morphology and relative observations configuration."

Reviewer's comments

I appreciate the authors' efforts in addressing my previous comments and revising the manuscript based on the initial feedback. While most of the authors' responses satisfactorily address my concerns, some aspects regarding the robustness of the methodology remain questionable, despite the additional sensitivity analysis provided. In this analysis, the authors examine how much the classification changes when noise is applied to four parameters — the convective fraction (F), convective area (A), characteristic length (L), and percentile (P). This is a valuable addition, as the manuscript presents a methodological approach. However, the analysis only assesses the method's sensitivity to errors in these parameters, without evaluating the typical magnitude of such errors in practice.

We agree with the reviewer (and have clearly expressed in the manuscript) that we only assessed the method's sensitivity to errors/uncertainty in these parameters. We also have clearly stated that evaluating the magnitude of such errors in practice is not an easy task. Were new insights from new research provide such an evaluation, our uncertainty propagation technique could easily be replicated to incorporate such new information.

We show next that i) our estimate of the scene area is consistent with the true cloud shield area, ii) our estimate of F is consistent with the true convective fraction (Response 1). We implement a geometry-free estimation of F (that is not without its own limitations) that we use to elaborate more on the possible uncertainty on estimating F and propagate it through the algorithm to quantify its effect on the classification, which is small (Response 2). We do this after clarifying a little misunderstanding on the propagation technique and the 10% threshold mentioned in the sensitivity analysis.

Based on the authors' response regarding the definition of the scene area, I am concerned that typical errors in F (and possibly P) could exceed the 10% threshold mentioned in the sensitivity analysis.

There is a small misunderstanding about the interpretation of the 10% threshold. As described in the manuscript, we have propagated uncertainties spanning + or - 1%,5%,10% etc. The analysis revealed a value ~10% , above which the final classification is significantly impacted. Below this value the classification hardly changes. In propagating the 10% uncertainty in the F variable, ALL the cloud shields are perturbed in our computations using a uniform distribution. So, the 10% value has to be interpreted as a threshold for the whole set of cloud shields and cannot be compared directly, with for instance uncertainty that would impact a subset of the cloud shield population. Said otherwise, the effect of having 2% of the systems out of 50000 with a 100% uncertainty is very different (much less) than having a uniformly distributed 2% uncertainty on the 50000 shields.

I encourage the authors to either provide evidence that this is not the case or revise their methodology for defining the scene area. I also have follow-up (more minor) comments regarding the interpretation of P and the sensitivity to the tracking algorithm. Please see my detailed comments below

We discuss the impact of the area selection on P (and L) and clarify the misunderstanding about the interpretation of P (Response 3) before closing on the influence of the tracking algorithm comment (Response 4).

Scene area

I thank the authors for their detailed explanation of how the scene area is defined for each case. To my understanding, the scene area corresponds to the smallest rectangle that fully encloses the DCS (with one pixel of padding) and is aligned with the radar swath grid. I see two potential sources of bias in this definition: (1) the shape of the DCS, particularly its deviation from a rectangular form, and (2) its orientation relative to the grid.

Non-rectangular DCSs that are oriented at approximately 45° relative to the satellite grid will result in significantly larger scene areas than those that are rectangular and aligned with the grid. For the method to yield a meaningful classification of convective element arrangements within DCSs, such orientation- and shape-related biases are undesirable.

We first introduce the distribution of the cloud shield in the eccentricity-orientation phase space diagram. The eccentricity is the classical way to characterize the shape of the DCS and is readily available in the TOOCAN outputs. It is used in the following to estimate the departure from a rectangular shape as highlighted by the reviewer. The lower the eccentricity the more elongated the DCS; DCS with eccentricity above 0.5 can be considered as quasi-circular and DCS with an eccentricity below 0.2 can be thought of quasi-linear systems (e.g., *Liu, C., and E. Zipser, 2013: Regional variation of morphology of organized convection in the tropics and subtropics. J. Geophys. Res. Atmos., 118, 453–466, https://doi.org/10.1029/2012JD018409.*). Given the growing uncertainty in estimating the eccentricity and the angle for quasi-circular objects, we restrict ourselves to the eccentricity below 0.5 for which both parameters can be estimated with confidence. The (relative) orientation is computed as the angle between the major axis of the DCS and the TRMM/GPM swath (between 0 and 45°).

The figure R1 below shows the distribution of systems in this phase space. In the lower range of the eccentricity, the systems are observed with a wide range of orientation yet local maxima are found in the upper range of the orientation distribution. Hardly no systems are found with eccentricity below 0.15 (as already shown in the previous round of reviews). The case study put forth by the reviewer corresponds to the local maxima at 40-45° and eccentricity ~0.3.

[Figure]

*Figure R1: 2D histogram representing the joint distribution of the relative orientation angle with respect to the TRMM/GPM swath against the aspect ratio*

Before exploring the distribution of F or the impact on P we focus first on the distribution of the cloud shield area (CSA) and its estimate through scene area (SA) using our rectangle-based method. Figure R2 shows the diagram between the actual cloud shield (CSA) and the algorithm estimate (SA). There is a strong relationship between the 2 variables with a $R^2$ around 0.8. The regression slope is 0.5 indicating that the algorithmic CSA is roughly twice as large as the true A, which is expected.

[Figure]

*Figure R2: 2D histogram representing the joint distribution of the real cloud shield area (CSA) against the algorithmic scene area (SA)*

To rule out any specific orientation-geometry bias in the SA estimate with respect to the true CSA value, we have computed the linear correlation between the two across the phase space. The data have been selected in boxes of eccentricity and orientation and only the cases with at least 100 points are used in the computations. Figure R3.a indicates that in most of the cases, the $R^2$ is high without any region-specific departure.

[Figure]

*Figure R3: 2D histogram representing the joint distribution of the aspect ratio against the relative orientation angle with respect to the TRMM/GPM swath for a) the linear regression coefficient $R^2$ and b) the slope of the linear regression fit between SA and CSA*

As shown in Figure R3.b, the slope of the regression does reveal regime specific values which are investigated next more quantitatively. The figure below shows the regression slope for a given eccentricity, for a very elongated and an elongated system (Ecc =0.2 and 0.4) along with the mean of all Ecc values below 0.5.

[Figure]

*Figure R4: Extraction of binned values of the slope (from Figure R3) against the orientation angle for three given Ecc (aspect ratio) values shown in colors.*

For the very elongated case, the slope does not change much with the orientation angle. Similarly for angles above 15° where most of the cases are found, the less elongated system

case shows not much change with the orientation angle. Below 15° there are few cases and a slight tendency for the slope to decrease with the angle.

**The lack of a significant dependency of the regression slope with the orientation angle tells that for a given eccentricity, the bias of the estimation of SA shows no strong relationship with the orientation angle ruling out a possible large effect on the method.**

While it is acceptable for F to be treated as an algorithmic variable representing the convective fraction, this remains valid only if there is a clear and consistent relationship between F and the actual convective fraction (i.e., larger F values should correspond to higher convective fractions).

We replicate below the same analysis for the F parameter. We plot below the joint distribution of the real convective fraction as a function of the algorithmic F for the full population of DCS.

[Figure]

*Figure R5: 2D histogram representing the joint distribution of the real convective fraction against the algorithmic convective fraction (F), the linear regression is shown with the white dashed line*

This plot indicates a clear and consistent relationship between the real convective fraction as a function of the algorithmic F with a strong correlation of $R^2$=0.71. As anticipated the real convective fraction is mostly larger than the estimate, or equivalently the algorithmic area is mostly smaller than the true area.

[Figure]

*Figure R6:* *2D histogram representing the joint distribution of the aspect ratio against the relative orientation angle with respect to the TRMM/GPM swath for a) the linear regression coefficient $R^2$ and b) the slope of the linear regression fit between the true convective fraction and the algorithmic convective fraction F*

To rule out any specific orientation-geometry bias in the F estimate with respect to the true F value, we have computed the linear correlation between the two across the phase space. The data have been selected in boxes of eccentricity and orientation and only the cases with at least 100 points are used in the computations. The figure above indicates that in most of the cases, the $R^2$ is rather high without any local bias. For the very rare cases with eccentricity of 0.2 and angle above 35 we observed a less strong correlation (although significant with $R^2 \sim 0.6$). The range of variation of the slope at a given eccentricity is small and following the lines of the arguments for SA, this rules out also in the case of F any large effect on the results.

Based on my understanding of the current methodology, I am concerned that this relationship may not hold given the current definition of scene area. To illustrate this point, consider a simplified example of an elliptical DCS (aspect ratio ≈ 1/3) observed by two radars with different grid orientations (cases A and B in the figure below). As it is the same DCS, the method should ideally yield similar convective fractions and classifications in both cases.

[Figure]

However, as shown, case B produces a significantly larger scene area than case A — a difference of approximately (30–22)/22 = 36%. This increase in scene area could lead to a decrease in the derived convective fraction by about 26%, potentially resulting in different classifications for the same physical system depending solely on its orientation relative to the radar grid (as suggested by the additional sensitivity analysis). Differences in DCS geometry would add on this orientation bias, making the convective fraction F less representative of the actual physical convective fraction.

This is opposed to assuming an algorithmic F. The two cases should not give similar values, they should give similar sensitivity across the population. Or formulated in another way, the uncertainty resulting from the estimation of F should not influence the results of the overall unsupervised classification but there is no need for various F estimates to be similar. In addition, as discussed in the introduction there is a little misunderstanding in interpreting the 10% threshold of our sensitivity analysis. Nevertheless, we have tried to explore alternative ways to compute F as one source of uncertainty in estimating F and in the next section we detail the effect on the overall classification.

A geometry-free F estimation is obtained by fitting the best rectangle using the cloud shield coordinates in a two-dimensional space free from the resolution. This is achieved by allowing rotation and non-integer width and height (in the pixel space). This permits the estimation free from the constraints of the radar grid. Therefore, it should be noted that finding the optimal rectangle this way is not always feasible, as one of the rectangle's dimensions will often exceed the radar swath width, creating 'information-free' areas. Nevertheless, keeping these limitations in mind, we have computed F this way for all the systems. The comparison between

the two estimates is shown below in the form of a relative difference computed as (F_new-F_old)/F_old.

[Figure]

*Figure R7: Histogram of the relative uncertainty over F between the two methods (in %). The black dashed line indicates the median of the distribution while the red one illustrates the mean*

The median of the distribution is 0%, and the mean is 4.57%, with a standard deviation of 16.5%. The distribution is also asymmetric. The configuration mentioned by the referee (high aspect ratios and orientations close to 45°) are found on the positive tail of the distribution. We also note that sometimes negative relative errors occur, i.e., a decrease in F. There is therefore no systematic bias of one method over another, with a slight tail effect on the high values. The influence of the most extreme cases, such as those cited above, is indeed visible in the high values in the relative error, but it remains limited by the effects of numbers when we return to the total distribution of F. We note that unlike what we did in our error propagation technique, the distribution is here non uniform.

We therefore calculated the following metrics characterizing the robustness of the final classification in the same way, but using the actual distribution (no more uniform assumption) for the relative error injection calculated previously, and obtained the following results: silhouette score: 0.3206; ARI: 91.1%; proportion of mismatch: 2.96% (i.e., 1 602 scenes over 54132). By plotting these values on the sensitivity study graph shown in the previous answer, which is reproduced below, we obtain a value of ~5% equivalent injected error, which is well below the 10% threshold we had set as the limit (see cyan diamond in the following figure).

[Figure]

*Figure R8: Same as Figure B3 (see the previous round of responses). The blue squares indicate the corresponding values for each of the indicators, replaced on the F curve (in red).*

In conclusion, assuming an alternative way to compute the area is giving a reasonable uncertainty distribution (without being better or worse), then only a few percent of mismatches are observed, further ruling out, quantitatively, any large effect of this geometry-orientation problem.

In addition, I recommend that the authors include a clear description of how the scene area is defined in the manuscript, as this aspect is critical to the methodology and may substantially influence the classification results.

We modified the L374-375 as follows: "A scene is defined as the rectangular area that encompasses the cloud shield identified by the TOOCAN algorithm, as illustrated by the black contour in Figure 3 (left)." ⇒ "A scene is defined as the rectangle that most closely frames the outline of the cloud shield identified by the TOOCAN algorithm, as illustrated by the black contour in Figure 3 (left). For the TOOCAN-Radar dataset, this rectangle must be oriented in the direction of the satellite's orbit within the TRMM/GPM swath geometry. It is not possible to extend beyond this geometry because radar information is missing there. When the data is available, a margin of 1 pixel is also used at the edges."

Finally, the limitations associated with the definition of scene area should be discussed in the "Limits of applicability" section.

We have added one paragraph in the form of a cautionary note in the mentioned section to summarize the discussion. See the response to the editor at the beginning of the document.

Figures:

Some figures still lack panel labels (letters) for reference. This is particularly the case for Figure 14, where letters are mentioned in the caption but do not appear in the figure itself. For clarity and consistency, I recommend adding panel letters (e.g., a, b, c, …) to all relevant figures, as these are preferable to references such as "left" and "right."

We have modified Fig. 14's caption accordingly.

Large variations in scene area would also influence the P metric, which is likely to be larger for case B than case A, possibly exceeding the 15% threshold reported in the sensitivity analysis.

Uncertainty in SA could influence not only F, but also P and L. There are no obvious reasons for P to be larger in case B than in case A. Indeed, P depends on SA and A (note that A is the only parameter independent of geometry), as well as its decomposition into elementary structures and L. Therefore, it is not possible to make a direct estimate using the given example.

I encourage the authors to either (1) provide evidence that typical errors in F and P do not exceed the thresholds discussed in Figure B3, or (2) revise their method to reduce these biases.

Following the previous discussion, the same lines of arguments can be put forth for P as we did for SA and F and we think that the geometry-orientation cannot yield to a large issue in P. But unlike for these variables there is no straightforward way to redo the computations. Indeed, to remove the dependency on the angles would require reprojecting the data in the non-regular grid before reprocessing the archive, changing the decomposition and the scene generation without guarantee to be free of problems.

Beyond the fact that it is important to interpret the 4 variables together and not apart, as already argued, we would like to emphasize the covariation among the variables. The classification would be influenced by the covariation of the 4 variables together. Given the strong relationship between algorithmic SA and the true SA, there are no obvious reasons to imagine that covariation in the 4D space is disturbed by the geometry-orientation. So, it is difficult to anticipate the amount of misclassification.

Also like we did for the F explanation, the fact that there is a strong consistency between the model-based results, free of geometry-orientation artefact, and the radar case indicates that if any effect there is, it is not large.

Interpretation of the percentile P:

> • L. 465-466 : "computing a percentile P, which quantifies the deviation of the real scene's spatial arrangement from a random arrangement (Figure 9, Red arrow in Figure 6)."

That wording implicitly assumes that higher percentiles mean stronger deviations from random.
But that is not necessarily true, low percentiles can also represent strong but opposite direction deviations (more regular patterns instead of random). Please reformulate.

> • L. 470-473 :

For clarity, I copy the Authors' response to my comment on the previous round of revision.

"We don't use the clustering notion to illustrate the meaning of the value of P. Besides, rarity is somehow a vague notion that can lead to ambiguous interpretations. We don't mention it directly. Instead, we express the notion of how many times an event occurs

within the total number of attempts in the context of the generated scenes, as the percentile computation. Besides, as described in the paper, our objective is broader than simply distinguishing between aggregated and clustered, as with metrics such as Iorg/Lorg, for example. Rather,we aim to quantify the deviation from randomness and the structuring of convective cores. Therefore, we believe that the original phrase is appropriate in this context."

I understand that the objective of the paper is broader than simply distinguishing between aggregated and clustered patterns, and the proposed method indeed achieves this, as indicated by the combination of P with the three other variables. However, my comment focuses specifically on the variable P.

The variable P measures the fraction of randomly generated scenes that have a lower L than the actual scene. Therefore, to my understanding, the variable P, as defined in the current version of the manuscript, is more of a measure of the degree of clustering expressed in a statistical sense (on a probabilistic scale) than a measure of the deviation from randomness. This is because both very low and very high values of P indicate strong deviation from randomness, as noted above. Conversely, as P increases from 0 to 1, the spatial arrangement transitions progressively from more regular to more clustered scenes. If the Authors would like to put more emphasis on the deviation from randomness than the degree of aggregation, I would suggest to use a modified metric, for example $1-2.\min(P,1-P)$, that will more closely quantify this aspect.

> • L.539-540 "On the other hand, the P distribution, indicating the probability of the characteristic length to differ from a random arrangement (Figure 12, d) is very much skewed to the highest values (around 1) with a very long tail. This indicates that only very few scenes can be associated with a randomly generated pattern."

The P distribution does not directly indicate a probability of differing from a random arrangement. First, very low P values are also indicative of deviation from randomness, as noted above. Second, such P-values (percentiles) only quantify the conditional probability of observing a value less extreme than the observed statistic (here L) under the null hypothesis (here, H0 = randomness (H1 = non-randomness)). Formally, they measure $\text{Prob}(L \le L_{obs} \mid H0)$, which has no direct relation with $P(H1 \mid L \le L_{obs}) = 1 - \text{Prob}(H0 \mid L \le L_{obs})$.

We agree and modified the wording on the incriminated sentence:
"On the other hand, the P distribution is very much skewed to the highest values (around 1) with a very long tail (Figure 12, d). This indicates that most of the scenes exhibit a characteristic length that cannot be easily reproduced by random generation".

To illustrate, consider a bag containing many dice — some fair, some modified. You randomly draw one die, roll it, and obtain a six. Under the null hypothesis of a fair die, the probability of rolling a six is 1/6, and thus the P-value is relatively high (not indicative of deviation). However, this outcome alone tells us nothing about whether the die was fair or not. The P-value only quantifies how likely the observation is if the die were fair; it does not give the probability that the die is unfair given the observation.

Similarly, in your analysis, the P metric does not directly represent the probability of nonrandomness.

While the mapping between P and the probability of deviation from randomness may appear more meaningful when the reference (random) distribution is approximately bell-shaped, there is no one-to-one correspondence. The interpretation of P thus depends both on the shape of the null distribution and, in a Bayesian sense, on the prior probability of non-randomness, which can vary across cases.

• L. 574 : "very unlikely to be randomly spatially distributed with high L values"

This statement remains somewhat ambiguous. It could be misinterpreted as referring to the unlikelihood of observing L relative to all high values in a random distribution, rather than the intended meaning: that the observed spatial pattern is specifically unlikely under the null hypothesis and falls on the side of clustered rather than regular patterns.

Again, we do not share the referee's vision around the interpretation of P. Figure R9 shows an example of a regularly spaced convective field where P has high value, illustrating that P alone is not particularly interpretable. It has to be compared to the 3 other key parameters to be understandable, as we state all along the manuscript. P is not comparable to an integrated metric like Iorg or Lorg, as we previously expressed in the manuscript and the previous round of response.

[Figure]

F=0.04 S=100 L=2.83 P=87.4

*Figure R9: Idealised case of regularly spaced convective field. The 4 key parameters are computed and shown on the top of the figure, with specifically low L and high P*

Sensitivity to the TOOCAN seed definition:

In the previous round of review, I shared my concern that the method used for detecting DCSs in the TOOCAN algorithm may restrain the diversity of spatial arrangements of DCCs, even when these DCCs are identified through radar measurements.

While the response provides a useful description of the TOOCAN procedure, it does not demonstrate that the IR-based identification of convective systems is fully independent of radar-based DCC detection (microwave measurements).

We do not understand this statement. We emphasized already that our study is TOOCAN-dependent and that we provide enough information for the interested reader to repeat our analysis using any other DCS identification algorithm.

Furthermore, it should be emphasized that any identification procedure (single threshold, Detect and Spread in 2D, or TOOCAN like approach) would identify the same features in the IR if the system is a well-defined, isolated large storm, like for instance the MCC of Maddox et al. (1980). For this case, the DCS delineation technique does not influence the detection much. TOOCAN departs from alternative techniques in the more complex cases where multiple systems do merge and split. It is also in this case that TOOCAN provides a more physically consistent depiction of the cloud shield than other techniques as shown in previous, already cited works. In particular the interested reader can find such examples based on idealized case studies described in Prein et al (2024).

Given that cloud-top temperature and convective precipitation intensity are often correlated (e.g., Arkin and Meissner, 1986),it seems surprising to me that the TOOCAN definition of DCS, which emerge from single convective seeds, would have no impact on the spatial arrangement of DCCs identified in radar.

We are very puzzled by this statement. The reviewer refers to a well-known work between accumulated surface precipitation and cold cloudiness area at the monthly and 250x250km scale to support his statement that at the instantaneous scale there is an impactful relationship between the TOOCAN seeds (cold IR temperature) and the deep convective cells (no surface precipitation).

The lack of correlation at short and small scales between IR cloud top, convection in column and surface precipitation has been well described in the literature and is the fundamental reason to use radar (or passive microwave) for surface precipitation estimate. For instance the seminal work of Yuter and Houze ( YUTER, S. E., and R. A. HOUZE JR, 1998: The natural variability of precipitating clouds over the western Pacific warm pool. Q. J. R. Meteorol. Soc., 124, 53–99, https://doi.org/10.1256/smsqj.54503.) addressed this issue in a quantitative manner. More generally, the interested reader can find some recent updates on the satellite-based estimation technique and how the limitations of the IR can be overcome in various review articles. A good entry point could also be the joint GEWEX/IPWG assessment report on satellite-based precipitation estimates (Roca et al., 2021).

The irrelevance of the first statement "given that" implies that the syllogism used by the reviewer in this comment is actually not valid.

Since convective seeds in the TOOCAN algorithm are defined as very cold brightness temperature minima, and only one seed is retained per DCS, one would expect the radar identified DCCs to be spatially close to these IR-based seeds. This makes it difficult to conceive how the TOOCAN DCS definition could have no influence on the spatial distribution of radar-detected DCCs.

We suggest that there is possibly a misunderstanding induced by the loose terminology used in the IR based world and the radar world. Convective seeds refer to cloudiness with a cold top while deep convective cores correspond to the column. In particular the TOOCAN seeds are defined over 3-time steps (1h30 in the present case) and with a minimum extension of 625km2 (Fiolleau and Roca 2013, Roca et al 2017, the present manuscript) which is not the same as instantaneous, km-scale convective cores depicted by the TRMM radar.

A brief quantitative check (e.g., distance statistics between TOOCAN seeds and radar DCC centroids), or reference to an earlier validation study would help clarify this point.

The radar observations are available only at the overpass time and not every time step of the life cycle of the DCS. So, statistics between TOOCAN seeds and radar DCC centroids makes no actual statistical sense.

Reference:
Arkin, P. A., and B. N. Meisner, 1987: The Relationship between Large-Scale Convective Rainfall and Cold Cloud over the Western Hemisphere during 1982-84. Mon. Wea. Rev., 115, 51–74, https://doi.org/10.1175/1520-0493(1987)115<0051:TRBLSC>2.0.CO;2.